# A Study on Customer Behavior in Online Dating Platforms: Analyzing the Impact of Perceived Value on Enhancing Customer Loyalty

**DOI:** 10.3390/bs14100973

**Published:** 2024-10-21

**Authors:** Qianghong Huang, Ru Zhang, Hyemin Lee, Huayuan Xu, Younghwan Pan

**Affiliations:** Department of Smart Experience Design, Graduate School of Techno Design, Kookmin University, Seoul 02707, Republic of Korea; huangqianghong@kookmin.ac.kr (Q.H.); 593838315@kookmin.ac.kr (R.Z.); hyeminest@kookmin.ac.kr (H.L.); 13256044989@163.com (H.X.)

**Keywords:** online dating platform, perceived value, satisfaction, customer loyalty

## Abstract

Customer loyalty is critical for organizations to gain market share and maintain a sustained competitive advantage. However, no study has yet explored customer loyalty in online dating platforms. Perceived value theory suggests that perceived value is a key predictor of customer loyalty. Accordingly, this research constructed a conceptual model drawing on a multidimensional perspective of perceived value to explore customer loyalty in online dating platforms and investigated the mediating role of satisfaction. By quantitatively analyzing 352 customers who had experienced online dating platforms and utilizing structural equation modeling (SEM) to examine the relationships among propositions, the research demonstrated a strong positive correlation between perceived value, satisfaction, and loyalty, and a notable indirect impact on loyalty through satisfaction. In addition, the experiential value dimension of perceived benefits had the most positive and substantial influence on perceived value, while the perceived risk dimension of perceived sacrifice had the most negative and notable impact on perceived value. The results of the study provide designers, managers, and vendors of online dating platforms with valuable insights into customer behavior and practical recommendations for improvement, helping them to develop more effective strategies to enhance market competitiveness and ensure the sustainability of their platforms.

## 1. Introduction

The internet has transformed people’s lifestyles, providing more opportunities for individuals to connect with others, expand their social networks, and find potential partners. Against this backdrop, online dating platforms have gained popularity among users [1]. These platforms can match users with prospective romantic partners and enable their initial interactions [2]. Reports indicate that nearly 40% of young individuals find their partners through online platforms, and by 2029, the global count of online dating users is projected to surpass 470 million [3]. As of June 2024, Tinder is the most downloaded dating app globally, with Bumble and Litmatch ranking second and third, respectively [4]. Despite its large customer base and rapid market growth, Tinder faces stiff competition in the market. As shown in Figure 1, there are several competitors offering similar services in the global market [4]. In the China market, Momo is the most popular dating app as of June 2024, but the company also faces fierce competition from many other competitors that provide similar services, such as TanTan, Baihe, iliao, Bumble, Qingteng Love, Soul, and other platforms [5]. In an increasingly competitive market environment, it has become common for customers to frequently switch between different platforms [6]. To meet this challenge, companies must not only strive to keep current customers, but also actively draw in new customers, with customer loyalty being vital to this process [7]. Loyal customers usually continue to use the company’s offerings, and through word-of-mouth communication, they can influence others, thereby helping companies acquire new customers at a lower cost [8,9]. Customer loyalty plays a key role in corporate profitability and sustainable development [10]. Loyal customers not only bring stable income to enterprises, but also help enterprises gain market share in competition and maintain long-term competitive advantages [11].

However, despite the fact that online dating platforms have a high prevalence, there are no specific studies explaining the antecedents of behavioral intentions to use such applications [12]. Research on online dating platforms has focused on risks of use [13,14,15], psychobehavior [16,17,18], homosexuality [19,20], sociodemographics [21,22], and motivations for use [23,24,25]. It is unclear which factors influence customer loyalty on online dating platforms, which is a research gap. As online dating continues to evolve, businesses need to determine the key factors affecting customer loyalty and take measures to optimize the customer experience so as to retain current customers, increase loyalty, and promote sustainable development of the business [9].

Perceived value serves as a crucial predictor of customer loyalty and post-purchase attitudes and holds significant importance in the economic growth, survival, and competitiveness of a business [26,27]. In marketing, one of the primary tasks of a business is to provide value to customers and effectively communicate this value so as to boost customer satisfaction and loyalty, thereby enhancing profitability [28]. Perceived value stems from the consumer’s overall assessment of the benefits and costs related to a product based on their evaluation of the product’s utility [29]. If consumers perceive that the product or service is good value for money after comparing the benefits and sacrifices, they will be satisfied, thereby positively influencing loyalty [11]. The perceived value theory has been widely used in customer loyalty research in many fields, such as retail [30], e-commerce [26], digital marketing [31], mobile payments [32], and online transportation [33]. However, online dating platforms have unique characteristics, and the perceived value theory has not been fully explored in these research fields. Online dating platforms not only provide convenience value (e.g., the freedom to communicate anytime, anywhere) [34], information value (e.g., learning about potential partners through personal profiles) [35], experience value (e.g., exploring emotional and entertainment experiences) [36], and social value (e.g., gaining recognition through interactions) [37], but also involve a large amount of customers’ private and personal information. Compared with other industries, customers’ concerns about privacy breaches, fraud, and other issues make perceived risk particularly prominent on this platform [14]. This sensitivity and the high relevance of risk is not commonly found in other research areas of perceived value. Therefore, applying the perceived value theory to online dating platforms, especially considering their unique risk factors and emotional interaction needs, is not only appropriate and necessary, but also effectively fills the research gaps in the existing literature.

Therefore, this study constructed a conceptual model to explore online dating platform customer loyalty based on the perceived value theory by quantitatively analyzing 352 customers who have experienced online dating platforms, and using structural equation modeling (SEM) to test the relationship between the hypotheses. The findings assist online dating platform designers, managers, or vendors in identifying the specific elements influencing customer loyalty and in optimizing the customer experience so as to bolster market competitiveness and enhance customer loyalty.

The remainder of this paper is structured as follows: Section 2 reviews the relevant literature on online dating platforms, perceived value, customer loyalty, and satisfaction. Section 3 proposes nine theoretical hypotheses to construct our research framework. Section 4 outlines the research methodology, encompassing questionnaire design, participant selection, data gathering procedures, and statistical analysis techniques. Section 5 provides the findings of the data analysis, covering reliability and validity analyses as well as hypothesis testing. Section 6 offers an analysis of the results. Section 7 summarizes the significance of the research, key conclusions, limitations, and directions for future research. The entire research process strictly complies with academic standards to guarantee the rationality, validity, and reproducibility of the methods used.

## 2. Literature Review

### 2.1. Online Dating Platforms

Online dating has been a significant focus of study in psychology and the social sciences [14]. Online dating platforms utilize technology to provide customers with the opportunity to find potential mates, facilitating a variety of relationships, including romance and marriage [15,38]. These applications provide an online space for users to set up profiles, look for prospective partners, and initiate conversations [12]. Customers can discover matches that align with their criteria, including factors like age, education level, profession, and hobbies, by using visual profiles and text messages [12,39]. Customers can express their liking by swiping right and disinterest by swiping left, and those who like each other can utilize the in-app chat function for further communication [40,41]. The emergence of these platforms has dramatically changed the traditional model of matchmaking and dating.

Currently, a variety of theoretical models have been used in studies on online dating platforms. For example, Patre et al. employed the elaboration likelihood model to understand customers’ dating intentions in a meta-universe [23]. Sowmya et al. used Satisfaction Theory to examine the factors affecting purchase motivation, adoption recommendations, and customer well-being on dating platforms [12]. Castro et al. utilized personality trait theory to examine the use of dating apps, sociodemographic factors, and the connection between light and dark personality traits [42]. Chakraborty et al. examined the determinants influencing the adoption of online dating platforms by utilizing the TAM and UTAUT frameworks [43]. Mosley et al. applied attachment theory to explore adult attachment, the developmental aspects of online dating, and the potential for deception [44]. Van Ouytsel et al. employed lifestyle–daily activity theory to explore online dating abuse victimization among middle school students [45]. Siddiqui et al., on the other hand, employed innovation resistance theory to explore the factors that impede online dating platform usage [38]. The survey of existing literature indicates that the perspective of perceived value theory has not yet been applied to explore customer loyalty on online dating platforms. Table 1 provides an overview of previous research on online dating platforms.

### 2.2. Perceived Value

Perceived value results from consumers’ assessment of a product, considering their holistic view of benefits and sacrifices, along with the assessment of the product’s utility [29]. It involves consumers making mental evaluations of costs and benefits through price comparisons during the process of buying products or services [48]. Specifically, perceived value can be understood as an equilibrium between perceived benefits and perceived costs, where the perceived value of an offering increases when customers believe the benefits exceed the costs [49]. The perceived value theory posits that perceived value is a key determinant of customer loyalty and post-consumption attitudes [11,26,50], playing an essential role in an organization’s economic development, survival, and competitive advantage [27,51]. Holbrook emphasizes that perceived value serves as “the core foundation of all marketing” [52]. Customers experiencing lower perceived value are more prone to turning to competitors, resulting in reduced loyalty [9].

In numerous studies on the concept of value, scholars have recognized the multidimensional nature of value, but a consensus has not been reached regarding how many dimensions exist or the classification standards [11,53]. Holbrook suggested that perceived value consists of eight dimensions, including efficiency, recreation, excellence, aesthetics, status, moral respect, and spirituality, with each of these dimensions being interconnected [54]. Hsu et al. categorized perceived value into four dimensions: informational value, experiential value, social value, and transaction value [55]. In the context of mobile applications, Jiang et al. classified perceived value into social, informational, and hedonic dimensions [56]. Singh et al. divided perceived value into convenience value, monetary value, social value, and emotional value in the context of streaming applications. Additionally, some scholars have suggested that perceived value can be distilled into two main dimensions: perceived benefit and perceived sacrifice. For instance, Bian et al. identified that perceived benefits encompass functional, social, emotional, cognitive, and experiential values, while perceived costs involve purchase expenses, purchase risks, and usage expenses [57]. Karsen et al. demonstrated that perceived risks and costs are crucial elements influencing the use of mobile applications [58,59]. Table 2 presents an overview of the prior research literature on perceived value.

Online dating platforms provide customers with a convenient platform to communicate with potential partners anytime and anywhere. The platforms display detailed information about a large number of potential partners, and customers can get to know each other without interaction [34,35,60]. In addition, customers can also pursue fantasy, emotional, and entertainment experiences on the platform [36], and gain recognition through active self-presentation and image building [37,61]. However, due to the lack of rigorous background checks and verification of personal information on the platform, fraud and illegal activities often occur [14]. At the same time, customers need to pay high membership subscription fees to use these services [62]. Based on the previous research, perceived value should include both benefits and sacrifices. Therefore, this study divides the benefits of perceived value into convenience value, information value, experience value, and social value, and the sacrifices into perceived risk and perceived cost.

**Table 2 behavsci-14-00973-t002:** Summary of prior research literature on perceived value.

Authors	Platform	Dimensions of Perceived Value	Ref.
Hsu et al. (2017)	Online game	Information value, experiential value, social value, transaction value.	[55]
Gan et al. (2017)	Social Business Platform	Utilitarian value, hedonic value, social value, perceived risk.	[63]
Oyedele et al. (2018)	Streaming apps	Convenience value, cognitive effort, emotional value, social value, monetary value.	[64]
Hamari et al. (2020)	Video game	Emotional value, social value, perceived quality, economic value.	[65]
Ashraf et al. (2021)	Mobile Business Platform	Information value, convenience value, monetary value, social value, performance value.	[66]
Singh et al. (2021)	Streaming apps	Convenience value, monetary value, social value, emotional value.	[67]
Jiang et al. (2022)	Mobile application	Social value, information value, hedonic value.	[56]
Wu et al. (2023)	Mobile Live Streaming Apps	Utilitarian value, hedonic value, social value.	[68]
Zhong et al. (2023)	Mobile Payment Platform	Functional value, experiential value, social value, perceived risk, perceived cost.	[11]
Dastane et al. (2023)	Mobile Business Platform	Practical value: information value, economic value, convenience value. Interaction value: interface value, visual value, gamification value, customization value. Trustworthiness value: system trustworthiness value, social trustworthiness value.	[69]
Zheng et al. (2024)	Educational applications	Utility value, hedonic value, social value.	[70]

### 2.3. Loyalty and Satisfaction

Customer loyalty refers to their positive disposition toward a brand [71]. Building and sustaining customer loyalty enables a company to establish enduring, reciprocally advantageous connections with its customers [7]. Loyal customers demonstrate attachment and commitment to the company, making them less likely to be swayed by competitors’ products [72]. As competition intensifies in the global marketplace, customer loyalty has become a key issue for companies to gain long-term competitive advantage [9]. Cultivating customer loyalty offers a substantial competitive advantage; loyal customers can not only be a source of financial income for a company, but also influence those in their network through recommendations and encouragement, allowing organizations to acquire additional customers at a reduced expense [8]. For application developers and service providers, establishing and sustaining long-term customer relationships is crucial for achieving sustainable growth [10]. Moreover, loyalty is crucial to resist brand switching [72]. In the global marketplace, online dating platform providers face intense competition; therefore, enhancing customer loyalty is critical for online dating platform providers to gain market share as well as to gain a sustainable competitive advantage [11]. In this research, customer loyalty denotes the intent of customers to continue using the current provider’s online dating platform and to recommend it to others.

Satisfaction is characterized as how a customer assesses the attributes of a product or offering [73], reflecting the cumulative feelings developed through multiple interactions with the service provider [74]. Satisfaction is considered an important factor in loyalty, which has been fully confirmed in previous studies of e-commerce platforms and mobile websites [75,76,77]. Customer satisfaction with the platform can enhance the user experience and consumer expectations, thereby positively influencing loyalty [11]. Furthermore, perceived value is recognized as the foremost and broadest antecedent of satisfaction [78], and the improvement of perceived value can directly improve overall customer satisfaction [79]. Perceived value is considered a crucial determinant of customer satisfaction [80]. Therefore, this study considers satisfaction as an important component of customer loyalty and explores its role in the online dating platform context and its relationship with perceived value and loyalty.

## 3. Hypothesis Development

This study, based on platform characteristics and drawing on the relevant research of Zhong et al. [11,66,67], divides perceived value into two components: perceived benefits and perceived sacrifices. Perceived benefits include four dimensions: convenience value, information value, experiential value, and social value, while perceived sacrifices consist of two aspects: perceived risk and perceived cost.

### 3.1. Perceived Benefits

#### 3.1.1. Convenience Value

Convenience value refers to the time and effort that customers save with a product or service so that they can accomplish the required tasks in a more efficient and appropriate manner [81]. Convenience ranks as one of the primary factors for individuals utilizing online platforms [82]. Online dating platforms offer a more convenient and effective means of connecting with potential dates, allowing customers to communicate without time and location constraints [34]. In a society where life is fast-paced, the convenience offered by online dating platforms has elicited a strong response [83]. Research by Singh et al. observed that convenience value positively influences customers’ perceived value [67].

Thus, we suggest the following hypothesis:

**H1a.** 
*Convenience value will positively influence perceived value.*


#### 3.1.2. Information Value

Sweeney et al. state that information value allows customers to access detailed information quickly and easily [84]. Customers’ expectations of knowledge, information, or resources on social platforms can help them access the information they need, and the value of this information can even help them create a competitive advantage [85]. Online dating platforms have a large amount of information about potential partners, often including information about age, tastes, education, and geography, and customers can learn about potential partners without interacting with the individual [35,60]. Research conducted by Lin et al. noted how consumers access information to fulfill their personal needs while on the move, which increases perceived value [86].

Thus, we suggest the following hypothesis:

**H1b.** 
*Information value will positively influence perceived value.*


#### 3.1.3. Experiential Value

Experiential value arises from the emotions or feelings experienced when using a mobile application [87]. Fang et al. argue that experiential value exerts the greatest influence on the willingness to adopt in the mobile domain and must be considered an essential element for mobile platforms to maintain an advantage in the market [88]. Customers seek experiences such as fantasies, sensations, and fun through the use of products or services, which can be achieved by using online dating platforms [36]. Hsu et al.’s study indicates that customer experience positively influences the perceived value of a product or offering [55].

Thus, we suggest the following hypothesis:

**H1c.** 
*Experiential value will positively influence perceived value.*


#### 3.1.4. Social Value

Social value pertains to the strengthening of social connections facilitated by mobile platforms [88]. If using mobile services enhances customers’ image and expands their social networks, they tend to keep using these services [89]. In recent years, social value has become increasingly significant in mobile services, with a growing number of mobile platforms heavily relying on social features and customer-generated content [65]. Customers can present themselves on these mobile platforms to effectively build and maintain relationships, seek companionship or verbal reinforcement, and pursue social recognition [90]. Online dating platform customers gain recognition by building their self-image and positive self-presentation [37,61]. Singh et al.’s study noted that social value exerts a beneficial influence on customers’ perceived value [64,67].

Thus, we suggest the following hypothesis:

**H1d.** 
*Social value will positively influence perceived value.*


### 3.2. Perceived Sacrifice

#### 3.2.1. Perceived Risk

According to Bauer’s definition, perceived risk is “the possibility that a consumer’s behavior may have outcomes which are beyond his anticipation, some of which may be unpleasant” [91]. Some researchers consider perceived risk as a sacrifice aspect of perceived value [11,86]. In our research, perceived risk denotes the likelihood and severity of negative outcomes resulting from the use of an online dating platform [14]. The matching algorithms used by online dating platforms do not merely collect vast quantities of personal information but also autonomously gather data like location, time, and previous activities [92,93]. Such a wide range of online sources exposes online dating platforms to higher privacy risks compared to traditional offline dating services [14]. Second, deception through online dating platforms is very common due to the lack of strict background checks and profile verification, and potential fraud and other illicit activities frequently take place on online dating platforms [94]. In an online dating platform environment, customers may also receive numerous unsolicited communications and requests that could result in disruption of their lives and psychological stress [14]. Aware of these risks, people may develop a negative attitude toward online dating platforms, which in turn leads to a decrease in the perceived value of their services [86].

Thus, we suggest the following hypothesis:

**H2a.** 
*Perceived risk will negatively influence perceived value.*


#### 3.2.2. Perceived Cost

Perceived cost denotes the degree to which an individual perceives that using a mobile service incurs expenses [95], and represents the amount the customer spends on a good or service and the sacrifices the consumer needs to undertake [96]. Verkijika et al. state that customers are more inclined to leave negative reviews if the transaction costs are elevated, unclear, or perceived to be inequitable [97]. In Lin et al.’s study, it was also shown that perceived costs exert an adverse impact on perceived value, and as the perceived cost to the consumer increases, their perceived value decreases, which can diminish the willingness to use mobile services [86]. Therefore, when there is a greater perceived cost to the consumer, this may lead to a lower perceived value of the online dating platform service.

Thus, we suggest the following hypothesis:

**H2b.** 
*Perceived cost will negatively influence perceived value.*


### 3.3. Perceived Value, Customer Satisfaction, and Customer Loyalty

A substantial body of research has confirmed a strong positive relationship among perceived value, satisfaction, and loyalty [11,98,99]. Perceived value serves as a pivotal factor in shaping customer satisfaction and loyalty within the context of mobile applications. To deter customers from switching to competitors, it is essential to continuously enhance perceived value [56,86]. Perceived value also signifies an emotional bond formed between the customer and the company after utilizing a product or service. This connection can offer additional value or gratification to the customer, and as the perceived worth and advantages increase, so do customer satisfaction and loyalty [100]. Studies have further discovered that customer satisfaction acts as a mediator between perceived value and customer loyalty [98,99]. Furthermore, customer satisfaction exerts a positive influence on loyalty [99,100].

Thus, we suggest the following hypotheses:

**H3.** 
*Perceived value will positively influence online dating platform loyalty.*


**H4.** 
*Perceived value will positively influence customer satisfaction.*


**H5.** 
*Customer satisfaction will positively influence online dating platform loyalty.*


### 3.4. Conceptual Model

Exploring the loyalty of online dating platform customers is important for the development, optimization, and promotion of the platform. This research develops a thorough model grounded in the multidimensional perspective of perceived value (Figure 2). According to the preceding discussion, the framework covers both the perceived benefits and perceived sacrifices components of perceived value, contains six dimensions, and considers satisfaction as an important component affecting loyalty. We aim to offer theoretical backing and empirical evidence for the design and marketing strategy of online dating platforms. Figure 2 illustrates the hypothesized relationships between the variables.

## 4. Research Methodology

### 4.1. Questionnaire Development

The gathering of questionnaire data comprised two components. The initial section gathered fundamental information regarding the customers, including gender, age, education level, and use of online dating platforms. The second section aligns with the research model and involves a total of nine variables: convenience value, information value, experiential value, social value, perceived risk, perceived cost, perceived value, satisfaction, and loyalty. All variables were derived from the previous literature to guarantee that the indicators were suitable for this study. Specifically, items for convenience value and information value were drawn from Ibáñez-Sánchez et al. [55,64,66,67,101]. Items of experiential and social value were sourced from the research by Zhong et al. [11,65,90,102]. Items of perceived risk and perceived cost were drawn from the study by Chen et al. [11,12,14,58,103,104]. Items for perceived value were sourced from the research by Singh et al. [67,86,102]. Items for satisfaction were derived from the research by Hsu et al. [12,105,106], while items for loyalty refer to Yuan et al.‘s study [11,101,106,107]. Each item was evaluated on a 7-point Likert scale, with options spanning from “1” for “strongly disagree” to “7” for “strongly agree”.

To ensure the accurate expression of the questionnaire, as the original questionnaire was in English and the study was conducted in China, we invited three professional English translation researchers to translate and proofread the questionnaire several times before it was distributed to reduce translation errors and eliminate ambiguities. Subsequently, we conducted a pretest of the questionnaire involving 30 customers with experience using online dating platforms to assess the logical flow and presentation of the questionnaire. Based on the researchers’ suggestions and feedback from the pretest, we modified the questionnaire to improve its clarity and structure. The 30 customers who participated in the pretest were excluded from the primary analysis to avoid potential bias arising from their prior exposure to the questionnaire. All respondents provided informed consent before completing the questionnaire to safeguard their rights and interests. Respondents were explicitly notified that the data collected would solely serve scholarly research purposes and that their privacy would be rigorously safeguarded. The list of entries and references for the questionnaire are provided in Table 3.

### 4.2. Data Collection and Participant Demographics

This study employs a cross-sectional online questionnaire design, with data collected through China’s professional online survey platform, Wenjuanxing (https://www.wjx.cn/, accessed on 7 July 2024). A questionnaire link and QR code were generated online, inviting users to complete the survey via social media platforms such as WeChat and QQ. Data collection utilized the snowball sampling technique across social media platforms [108]. Participants in the online questionnaire had the opportunity to enter a lottery, with prizes including (1) a 5 RMB WeChat red packet; (2) a 10 RMB WeChat red packet; and (3) a thank-you note for participation. In order to address the limitations of the snowball sampling method, we initially shared the questionnaire link within WeChat and QQ groups, avoiding an exclusive dependence on individual referrals. To ensure a more diverse sample, respondents were also requested to invite participants from various backgrounds, thereby reducing the risk of the sample being skewed toward a specific demographic.

All participants had experience in using online dating platforms. Throughout the process, participants participated voluntarily and there were no conflicts of interest. The researcher clearly communicated the study’s purpose to the participants and emphasized the principle of data confidentiality. Participants were also informed that they could stop participating at any time if they felt uncomfortable. In this study, a total of 378 questionnaires were collected. All responses were meticulously reviewed, and invalid entries were excluded according to the following criteria: (1) answering all questions exactly the same; (2) completing the questionnaire in an excessively short period of time; and (3) questionnaires with apparently contradictory answers. Twenty-six questionnaires were finally determined to be invalid. Thus, 352 valid questionnaires were collected for this research. Among the respondents, 48% were male, while 52% were female. The respondents’ ages were primarily concentrated in the 18–34-year-old range (84.1%). Of the respondents, 62.6% had used online dating platforms for more than 6 months, and 64.2% of the respondents had used online dating platforms for more than 1 h per day. These findings suggest that most respondents are inclined to use online dating platforms and are prepared to invest additional time. The descriptive analysis of the demographic information is shown in Table 4.

## 5. Data Analysis

According to Bollen, structural equation modeling (SEM) is a powerful second-generation multivariate analytic causal modeling technique used to estimate the two components of a causal model: the measurement and structural models [109]. Measurement modeling employs validated latent variable analysis to evaluate the constructs’ reliability and validity, while path modeling examines the significance and directionality of the relationships among the constructs. Therefore, we chose SEM as the method to analyze the collected data.

Given that all variables were collected through a single questionnaire, common method variance (CMV) is a potential concern, making it essential to test for CMV as the initial step of statistical analysis. Harman’s single-factor test was utilized to evaluate the CMV, and a principal component analysis was performed on all the items involved in the model using IBM SPSS Statistics 27.0.1. The analysis identified nine factors, with the first factor accounting for 11.6% of the variance, which is below 50%. Thus, the sample data from this study do not exhibit common method variance [110].

### 5.1. Measurement Modeling

#### 5.1.1. Measurement Model Fit Indices

To further test the construct validity of the model, a factor analysis was performed utilizing the structural equation modeling software AMOS 24.0. Hair et al. (1998) recommended that the majority of model fit metrics must meet acceptable criteria prior to making judgments on model fit [111]. According to the results of the measurement model fit test shown in Table 5, all the model fit metrics indicate that the model has an adequate fit to the collected data and exceeds the recommended values. Specifically, the CMIN/DF (chi-square degrees of freedom ratio) was 1.555, which fits within the range of 1 to 3, and the RMSEA was 0.04, which is in the excellent range of <0.05 [112]. In addition, the test results of the TLI and CFI both exceeded 0.9, which is an excellent level [113]. These results indicate that the measurement model has a good fit.

#### 5.1.2. Reliability and Validity Test

The questionnaire’s reliability and validity were evaluated through tests for internal consistency reliability, convergent validity, and discriminant validity. Internal consistency reliability was measured by calculating Cronbach’s α, Composite Reliability (CR), and Average Variance Extracted (AVE). As shown in Table 6, both Cronbach’s α and CR values surpassed the recommended threshold of 0.7, indicating satisfactory reliability [114,115]. Convergent validity was assessed by analyzing the factor loadings of each item within the measurement model, the significance levels of these loadings, the construct reliability, and the AVE. As presented in Table 6, all item factor loadings within the measurement model were above 0.7, and both AVE and CR values exceeded the 0.5 and 0.7 benchmarks, respectively, demonstrating that the scale possesses strong convergent validity [113]. In summary, all the constructs in the model have sufficient reliability and convergent validity.

Discriminant validity was evaluated based on the criteria set forth by Fornell and Larcker, ensuring that the square root of the AVE for each construct was greater than the correlations between any pair of distinct constructs [114]. As shown in Table 7, the results confirm sufficient discriminant validity. The Heterotrait/Monotrait (HTMT) ratio was used as the second criterion to confirm discriminant validity. According to the study by Henseler et al., all HTMT ratios were below the threshold of 0.85, indicating no issues with discriminant validity (Table 8) [116]. Consequently, we can conclude that these constructs demonstrate appropriate reliability and validity.

### 5.2. Assessment of Structural Models and Assumptions

#### 5.2.1. Structural Model Fit Indices

The research model’s path coefficients were evaluated through the use of AMOS 24.0. Based on the results of the structural model fit test in Table 5, it can be seen that the CMIN/DF (Cardinality Degree of Freedom Ratio) is 1.716, which fits within the range of 1 to 3, indicating a good model fit. The RMSEA (Root Mean Square of Error) is 0.045, which is in the excellent range of <0.05 [112]. In addition, the test results of the TLI (Tucker–Lewis Index) and CFI (Comparative Fit Index) both exceeded 0.9, which is an excellent level [113]. Therefore, synthesizing the results of this analysis, it can be shown that this model has a good fit.

#### 5.2.2. Hypothesis Testing

In this analysis, the path coefficients and p-values were calculated using a sample of 352 participants (*** *p* < 0.001; ** *p* < 0.01; * *p* < 0.05). According to the display in Table 9 and Figure 3, four dimensions of perceived benefits were significantly and positively correlated with perceived value in online dating platforms: convenience value (β = 0.140, *p* < 0.01), informational value (β = 0.138, *p* < 0.01), experiential value (β = 0.273, *p* < 0.001), and social value (β = 0.145, *p* < 0.01), and these results support hypotheses H1a-d.

Of the two dimensions of perceived sacrifice, perceived risk (β = −0.182, *p* < 0.001) was significantly and negatively correlated with perceived value, supporting hypothesis H2a. However, the link between perceived cost (β = −0.076, *p* = 0.149) and perceived value was not significant, rejecting hypothesis H2b.

In addition, perceived value significantly influenced both satisfaction (β = 0.635, *p* < 0.001) and loyalty (β = 0.320, *p* < 0.001), while satisfaction also had a significant impact on loyalty (β = 0.350, *p* < 0.001), supporting hypotheses H3, H4, and H5. Thus, the model was validated (Figure 3).

#### 5.2.3. Mediating Effects Test

In this study, the mediating role of satisfaction was evaluated through the bootstrap method in AMOS 24.0. This is a nonparametric method that can accurately test indirect effects [117]. The above information remains valid even with a limited sample size [118]. The mediating effect of satisfaction was supported by generating 5000 random observations from the original sample and obtaining estimates within 95% confidence intervals that do not include 0 (0.134, 0.321), indicating that perceived value has a significant indirect effect on loyalty through satisfaction. The amount of mediating effect is typically characterized as the decrease in the influence of the initial variable on the outcome, or as the difference between the total effect and the direct effect (i.e., |total effect − direct effect|). Another way to describe the mediating effect is by expressing it as the proportion of the indirect effect to the total effect (i.e., |indirect effect/total effect|) [117,119]. Thus, approximately 41% of the total influence of perceived value on loyalty is mediated by satisfaction. The total effect of perceived value on customer loyalty is 0.542, while the direct and total effect of customer satisfaction on customer loyalty is 0.35. Consequently, the total effect of perceived value on customer loyalty is stronger than that of customer satisfaction. In Table 10, we can observe the standardized total effect, the direct effect, and the indirect effect.

## 6. Discussion

Building on the multidimensional view of perceived value, this study explores the determinants of customer loyalty on online dating platforms, including both the gain and sacrifice components of perceived value. The findings of the study confirm the robustness of the model, while structural equation modeling (SEM) verifies its validity (Figure 4).

First, we observed a strong positive relationship between perceived value, customer satisfaction, and loyalty among online dating platform customers, aligning with the findings of previous studies [98,120]. Among them, perceived value exerts the strongest influence on customer loyalty, which corresponds with the results of Zhong et al., who observed in their research that perceived value serves as a critical predictor of customer loyalty [11,26]. Therefore, to achieve greater customer loyalty, online dating platforms need to continually strive to enhance consumer perceived value. Research has also demonstrated that perceived value has a significant effect on customer satisfaction, and Pang et al. observed in an earlier study that perceived value strongly impacts customer satisfaction, with increasing perceived value directly enhancing overall customer satisfaction [121,122]. In addition, customer satisfaction serves as a mediator between perceived value and customer loyalty, a result consistent with previous studies [80,99]. These findings suggest that designers, managers, and vendors of online dating platforms should focus on improving customers’ perceived value to enhance customer satisfaction and loyalty. It is worth noting that perceived value indirectly influences customer loyalty via satisfaction; therefore, enhancing perceived value to increase satisfaction and maintain customer loyalty is also an effective approach.

Second, this research explored the relationship among the six dimensions of perceived benefits, perceived sacrifices, and perceived value. The findings suggest that all components of perceived gain (convenience value, information value, experiential value, and social value) significantly influence perceived value, aligning with the findings of previous studies [11,67,68,123]. Notably, experiential value had the greatest impact. Fang et al. also highlighted in their study that experiential value is crucial for mobile platforms to sustain a competitive advantage [88]. This suggests that experiential value is a prioritized motivation for customers of online dating platforms. This implies that a consistently high level of product and service quality has ceased to be a major factor in consumer choice [124]. Within the growing experiential economy, customers do not merely pay attention to the provider’s offerings, but also to the memorable experiences and feelings during the usage process. Platforms can add a variety of interactive features, such as integrating entertainment content in the platform, such as games, live streaming, virtual gifts, etc., to make customer interactions on the platform richer and more interesting, and to increase the length of stay and stickiness of customers on the platform. Continuous innovation, such as the introduction of AR and VR technology, allows customers to experience dating in virtual reality, to enhance the customer’s interactive experience.

Social value has a positive impact on perceived value, which is consistent with the findings of Wang et al. [67,125]. As a dating and socializing application, online dating platforms uniquely combine the features of social media and matchmaking services. Users often seek to enhance their social image through these platforms, thereby achieving better outcomes in social interactions and dating experiences. To this end, future platforms can organize online or offline social events and themed activities, providing users with more opportunities to meet new people and engage in social interactions. By participating in discussions or events, users can gain more opportunities for self-presentation and recognition from others. Additionally, the platform could introduce a user rating or review system, where users earn points or titles through positive interactions, incentivizing them to maintain proactive behaviors. This approach would not only increase user engagement but also further enhance the platform’s social value.

Although convenience value has a positive impact on perceived value, it is not the most significant factor. This differs from the findings of Singh et al., who concluded that convenience value plays a crucial role in influencing customers’ perceived value and has the greatest impact [67]. This difference may be due to changing times, where convenience is no longer a key factor in attracting customers but has instead become a basic requirement for online dating platforms. Additionally, information value has a significant impact on perceived value, indicating that relevant, sufficient, accurate, and timely information can enhance customers’ perceived value and encourage them to use online dating platforms. Therefore, in the future, providers should continue to improve platform convenience by optimizing user interfaces and enhancing the overall user experience. This includes ensuring a simple and user-friendly interface design that reduces operational complexity and allows users to communicate anytime, anywhere. Personalized recommendations and smarter matching algorithms can help users quickly find potential partners. Platforms should also ensure seamless functionality across devices (mobile phones, tablets, computers), allowing customers to easily switch between devices. Moreover, users should be encouraged to provide more detailed and accurate personal information. Enhancing the quality of information on the platform will enable customers to better understand potential matches without needing direct interaction.

Finally, perceived risk exerts a significant adverse effect on perceived value, a finding that aligns with Chen et al.’s conclusion that perceived risk significantly reduces customers’ perceived value of online dating platforms [14]. This is because online dating platforms explicitly require customers to provide their cell phone numbers, personal photos, age, education, and other personal, private information, which leads to customers’ concerns about the risks associated with privacy leakage and being scammed, and therefore reduces perceived value [12,39]. While the role of perceived cost is relatively weak and the result is not significant, this outcome aligns with the findings of Zhong et al. [11,64]. It may be due to the extensive use of smartphones and the advancement of infrastructure in recent years, which have significantly lowered the cost for customers to use online dating platforms, coupled with the fact that the competition in the online dating market is relatively fierce, and expenses such as access fees and membership charges are relatively small, and the perceived costs have a smaller influence on the perceived value of online dating platforms to some degree compared with customers in other mobile application markets. Therefore, platforms should actively encourage and ensure customer authenticity and integrity to reduce false information and fraudulent activities. By implementing strict customer verification and review mechanisms, platforms can ensure the authenticity and reliability of user profiles. To reduce perceived risk, platforms should strengthen privacy and data security protection measures, enhance user identity verification, and use encryption technologies to boost customers’ trust in the platform. Additionally, improving the complaint and reporting mechanisms is essential to ensure that users can quickly report issues, and the platform can respond promptly to safeguard user rights.

## 7. Implications, Conclusions, and Limitations

### 7.1. Implications

The theoretical contributions of this research are primarily demonstrated through the following aspects: First, this study is the first to apply the perceived value theory to the study of customer loyalty in online dating platforms, although the theory has been widely used in customer loyalty research in other fields [32,120,126]. Second, this study combines the perceived value theory with the relevant literature [11,66,67] and divides perceived value into six dimensions based on the characteristics of online dating platforms: convenience value, information value, experience value, social value, perceived risk, and perceived cost. By constructing a multidimensional framework of customer loyalty, this research uncovers the significant influence of perceived value on customer satisfaction and loyalty, further expanding the theoretical foundation of this field. In addition, the results of this study provide scholars with a direction for future research, showing that perceived value not only significantly affects customer satisfaction and loyalty, but also indirectly affects loyalty through satisfaction. Among them, experiential value, social value, informational value, convenience value, and perceived risk are important dimensions that affect perceived value, which provides an important breakthrough for future theoretical research on online dating platforms. In summary, this research not only offers an innovative theoretical framework for examining online dating platforms, but additionally establishes a robust basis for upcoming theoretical investigations.

Attracting new users, retaining old users, and maintaining customer loyalty are crucial for online dating platforms [7]. From a practical perspective, this research offers several key suggestions for designers, administrators, and suppliers of online dating platforms. First, this study confirms the key role of perceived value in customer loyalty. Platform suppliers should continuously improve customers’ perceived value to deter users from migrating to rival platforms and achieve sustainable development. This is consistent with previous research [11,26]. Second, this study explores the impact of each value dimension on total perceived value, identifies the value components that are most important to customers, and helps platform designers develop effective strategies. Among them, experience value is the most valued dimension by customers, and earlier research has also indicated that experience value is crucial for platforms to maintain a competitive advantage [88]. As the social economy develops, individuals are paying more and more attention to meaningful experiences [127]. Therefore, platform providers should pay more attention to experience value and increase entertainment functions (such as games, live broadcasts, virtual gifts, etc.) to enhance customer loyalty. Simultaneously, perceived risk is also an important factor affecting customer loyalty. Platform designers and managers must comply with policies and regulations and implement effective information verification and privacy protection measures. By strengthening user authentication and information encryption, customer concerns about privacy leaks and fraud can be alleviated [14]. In summary, this study provides future directions for improving the practices of online dating platforms. It emphasizes the importance of improving customer experience and reducing perceived risk, which will help platform designers and managers develop more effective operational strategies.

### 7.2. Conclusions

This study, through the evaluation of a multidimensional model, reveals the significant influence of perceived value on customer satisfaction and loyalty, and highlights the indirect impact of satisfaction on customer loyalty. The results suggest that experiential value, social value, convenience value, information value, and perceived risk have significant effects on perceived value. Among these, experiential value emerges as the most positive and critical factor, suggesting a shift in customer demand from basic functional needs to personalized and enriched experiences. This underscores the need for platform designers to enhance user experience value through innovative interaction design, engaging content, and immersive experiences, thereby increasing platform attractiveness and competitiveness. Conversely, perceived risk is identified as the most significant negative factor affecting perceived value, reflecting customer concerns about privacy breaches and fraud. Platform providers must not only ensure functional security but also establish robust privacy protection mechanisms and respond promptly to customer complaints. These actions will foster customer trust, reduce perceived risk, and ultimately improve customer loyalty. In conclusion, to achieve higher customer loyalty and promote the sustainable development of online dating platforms, platform designers, managers, and providers must identify and prioritize the key factors influencing perceived value. By continuously optimizing platform features, enhancing user interaction experiences, and safeguarding privacy and security, they can sustainably improve overall perceived value.

### 7.3. Limitations and Future Research Directions

This research offers fresh perspectives on the factors affecting customer loyalty to online dating platforms through the perceived value theory, but there are still some limitations. First, this study relies on self-disclosed information gathered through surveys, potentially leading to response biases and social desirability biases that impact the accuracy of the findings. Subsequent research could incorporate quantitative surveys and qualitative methods (such as interviews or focus groups) to gain a more holistic and deeper understanding through mixed methods [12]. Second, this study’s sample was limited to 352 customers, and the restricted sample size, along with the geographical scope, could influence the general applicability of the results. Subsequent studies ought to expand the sample size and increase geographical coverage to improve the comprehensiveness and representativeness of the research results [128]. In addition, due to resource and time constraints, this study used cross-sectional data, which partially limits the ability to monitor and interpret long-term changes in customer behavior and motivation on online dating platforms [129]. Subsequent research might expand our findings using a longitudinal approach to offer a more thorough understanding of the dynamic shifts in customer behavior. Finally, although the multidimensional model of perceived value theory is widely acknowledged as an effective tool, it may not sufficiently cover all the factors influencing customer loyalty to online dating platforms. Therefore, future research could consider combining other theoretical frameworks or introducing more variables, such as attachment theory and social influence theory, to more comprehensively understand the behavior of online dating platform customers [130]. This will help to further refine and expand on existing research results.

## Figures and Tables

**Figure 1 behavsci-14-00973-f001:**
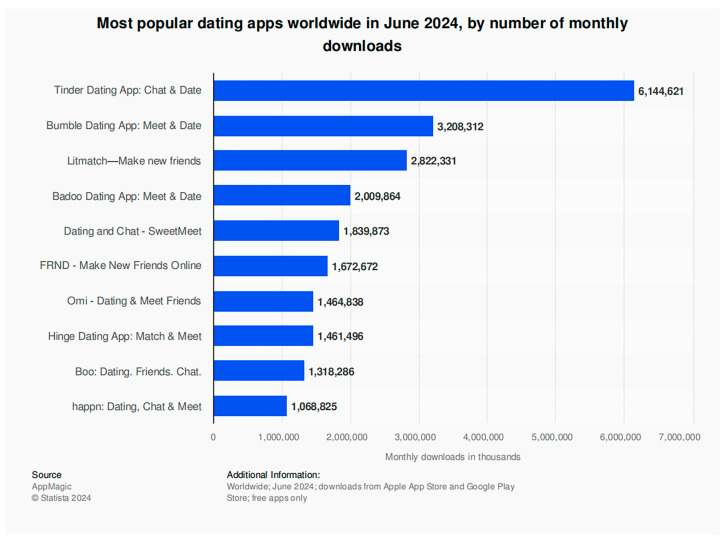
Most downloaded dating apps worldwide in June 2024 [4].

**Figure 2 behavsci-14-00973-f002:**
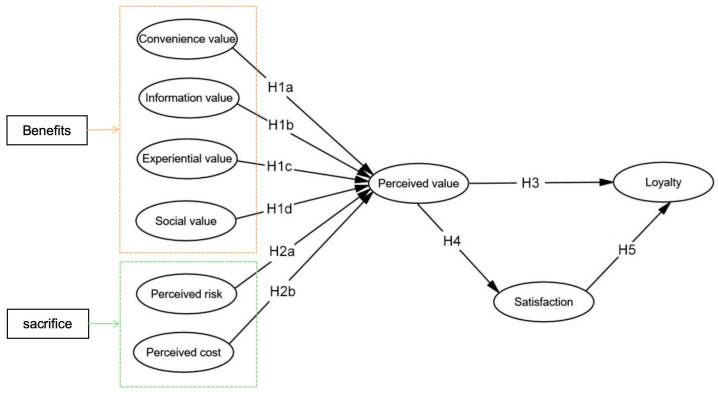
Proposed conceptual model.

**Figure 3 behavsci-14-00973-f003:**
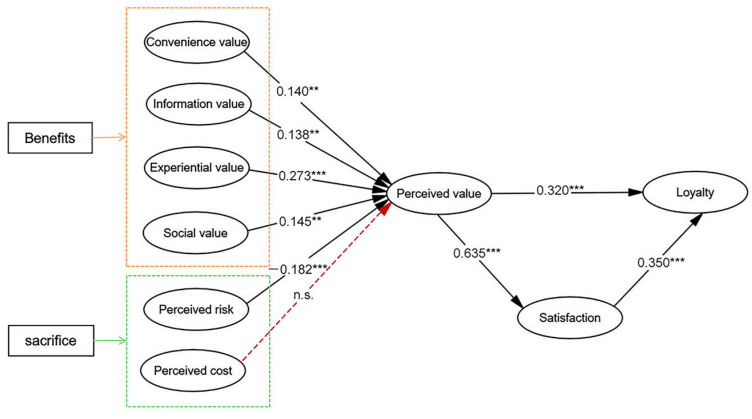
Model estimation results (note: *** *p* < 0.001; ** *p* < 0.01; n.s.: not significant).

**Figure 4 behavsci-14-00973-f004:**
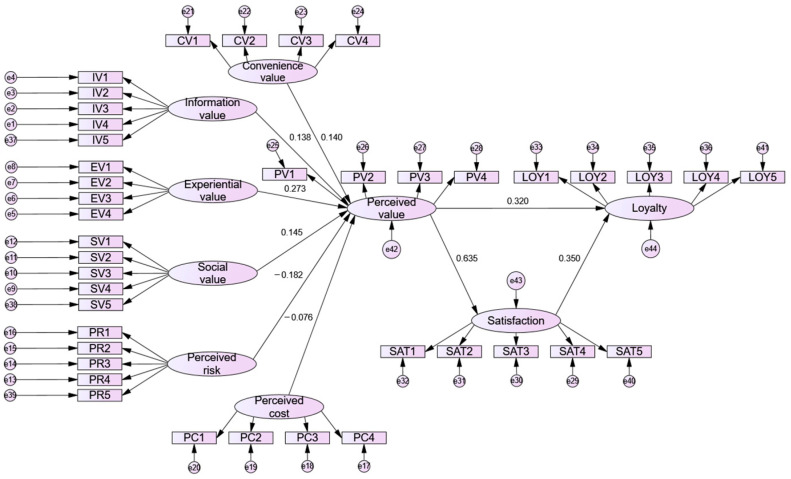
Results of SEM structural model.

**Table 1 behavsci-14-00973-t001:** Review of the research literature on online dating platforms.

Authors	Model/Theories	Purpose	Findings	Ref.
Van Ouytsel et al. (2018)	Lifestyle–routine activities theory	Victimization of secondary school students through online dating abuse.	Engagement in online risk behaviors, the duration of romantic relationships, participation in sexting with a romantic partner, and the frequency of social networking site usage were all significantly associated with being victimized by digital controlling behaviors.	[45]
Chakraborty et al. (2019)	Unified theory of acceptance and use of technology and trust	Factors affecting the adoption of online dating platforms.	Playfulness is the most influential factor, while trust has no influence at all.	[43]
Castro et al. (2020)	Personality traits theory	Analyze the relationship between the use of dating apps, sociodemographic characteristics, and bright and dark personality traits.	Men, older youth, and sexual minorities were more likely to be current or past dating app users, with being single and having higher open-mindedness scores further increasing the likelihood of current app usage, while dark personality traits showed no predictive ability.	[42]
Mosley et al. (2020)	Attachment theory	Understand the field of adult attachment, online dating development, and the possibility of deception.	Women are more likely to be targets, whereas men are more likely to be the perpetrators of this form of online dating deception.	[44]
Kaakinen et al. (2021)	Lifestyle exposure theory	Youth use of online dating platforms and victimization experiences.	Victimization is more common among young people using online dating applications, which are linked to increased risks of online harassment, online sexual harassment, cybercrimes, and sexual victimization by both adults and peers.	[46]
Tsui et al. (2022)	Social exchange theory	Understand and study the relationship between online dating platforms, relationship satisfaction, and self-esteem.	The relationships between users’ feelings on online dating apps, interpersonal relationship satisfaction, and self-esteem were significantly positive. The low cost of using these apps helps users achieve better satisfaction in relationships and higher self-esteem.	[47]
Patre et al. (2023)	Elaboration likelihood model	Investigate dating-partner behavior in the metaverse.	Interaction quality, avatar customization, visual attractiveness, intrinsic enjoyment, and perceived entertainment value are significantly correlated with perceived attitude.	[23]
Siddiqui et al. (2023)	Innovation resistance theory	Factors hindering the use of online dating platforms.	The three barriers of risk, usage, and tradition have a significant negative influence on the adoption intention of online dating apps.	[38]
Sowmya et al. (2024)	Uses and gratifications (U and G) theory	Factors influencing purchase motivation, adoption recommendations, and customer happiness in dating apps.	Convenience gratification, information gratification, and personal identity gratification are important predictors of purchase motivation for dating apps.	[12]

**Table 3 behavsci-14-00973-t003:** Questionnaire Items.

Variables	Items	Ref.
Convenience value	I think using a dating platform makes it clear and simple to get what I want.	[64,66,67]
	I find it convenient to use a dating platform to get what I want.	
	I think using a dating platform helps me save time and energy.	
	I find it convenient to use dating platforms compared to traditional dating methods.	
Information value	I think dating platforms offer me more than enough information.	[55,66,101]
	I think it’s cheaper to get information from dating platforms.	
	I think dating platforms provide information that meets my needs.	
	I think dating platforms offer me up-to-date information about potential dates.	
	I find the information I get from dating platforms very helpful in finding potential matches.	
Experiential value	I found the experience of using the dating platform to be enjoyable.	[11,90,102]
	I think using a dating platform gives me great pleasure.	
	I think it’s enjoyable to use a dating platform.	
	I love using dating platforms.	
Social value	I think that using a dating platform enables you to meet more people.	[11,65,90]
	I think using a dating platform allows for better communication with potential dates.	
	I can find out the status of other clients through the dating platform.	
	My friend thinks that using a dating platform to find a date is a good way to go.	
	I think using a dating platform will give me social acceptance.	
Perceived risk	I believe there’s a potential risk of my personal information being compromised by using a dating platform.	[11,12,14]
	I think using dating platforms makes me feel unsafe.	
	I think using a dating platform could lead to me being scammed.	
	I think using a dating platform may create negative rumors in my social circle.	
	Overall, I think using a dating platform is risky.	
Perceived cost	I think the cost of membership for accessing a dating platform is expensive.	[58,103,104]
	I think the additional fees for using dating apps are very high.	
	I think the investment cost of using dating apps would be very high.	
	Overall, I think using a dating platform would be quite expensive.	
Perceived value	I think online dating platforms are worth every penny.	[67,86,102]
	Considering the effort I invest, using an online dating platform is worth it for me.	
	Compared to the time I spent, using an online dating platform was worth it for me.	
	Overall, using an online dating platform has been valuable to me.	
Satisfaction	I think the dating platform met my expectations.	[12,105,106]
	I think using a dating platform makes me very happy.	
	My experience with the dating platform exceeded my expectations.	
	Using online dating platforms makes me feel very fulfilled.	
	Overall, I am satisfied with my experience on the online dating platform.	
Loyalty	I would make dating platforms my first choice for the way I find dates.	[11,101,106,107]
	I would recommend others to use the dating platform.	
	Changing my opinion of dating platforms is going to be very difficult.	
	My preference for dating platforms doesn’t change even if close friends recommend other dating personals.	
	I will continue to use dating platforms if I need to find someone again.	

**Table 4 behavsci-14-00973-t004:** Participant demographics (N = 352).

Variable	Items	Frequency	Percentage %
Gender	Male	169	48
	Female	183	52
Age (years)	18–24	100	28.4
	25–29	149	42.3
	30–34	47	13.4
	35–39	25	7.1
	40–44	14	4
	45 and over	17	4.8
Education level	Junior high school and below	17	4.8
	High school/secondary school	33	9.4
	Undergraduate/specialized	231	65.6
	Graduate students and above	71	20.2
Experiences with online dating platforms	Less than 3 months	54	15.3
	Up to 3–6 months	78	22.2
	Up to 6–12 months	82	23.3
	Within 1–3 years	47	13.4
	More than 3 years	91	25.9
Daily frequency	Less than 1 h	126	35.8
	Within 1–5 h	83	23.6
	Within 5–10 h	75	21.3
	10 h and above	68	19.3

**Table 5 behavsci-14-00973-t005:** Fit indices for measurement and structural models.

Fit Indices	Recommended Value	Measurement Model	Structural Model
Absolute Fit			
CMIN/DF	≤3.0	1.555	1.716
RMSEA	≤0.08	0.040	0.045
IFI	≥0.9	0.981	0.975
TLI	≥0.9	0.979	0.973
GFI	≥0.8	0.866	0.853
AGFI	≥0.8	0.845	0.833
Incremental Fit			
CFI	≥0.9	0.981	0.975
NFI	≥0.9	0.949	0.943
RFI	≥0.9	0.944	0.938
Parsimonious Fit			
PNFI	>0.5	0.860	0.868
PGFI	>0.5	0.747	0.748

**Table 6 behavsci-14-00973-t006:** Reliability and validity analysis.

Variables	Items	Factor Loadings	AVE	CR	Cronbach’s α
**Convenience value**	CV1	0.971	0.8715	0.9644	0.963
	CV2	0.907			
	CV3	0.928			
	CV4	0.927			
**Information value**	IV1	0.982	0.8731	0.9717	0.972
	IV2	0.917			
	IV3	0.918			
	IV4	0.906			
	IV5	0.947			
**Experiential value**	EV1	0.972	0.8894	0.9698	0.97
	EV2	0.926			
	EV3	0.912			
	EV4	0.961			
**Social value**	SV1	0.979	0.8868	0.9751	0.975
	SV2	0.920			
	SV3	0.933			
	SV4	0.918			
	SV5	0.957			
**Perceived risk**	PR1	0.978	0.8767	0.9726	0.973
	PR2	0.908			
	PR3	0.907			
	PR4	0.915			
	PR5	0.971			
**Perceived cost**	PC1	0.966	0.8648	0.9623	0.962
	PC2	0.906			
	PC3	0.883			
	PC4	0.962			
**Perceived value**	PV1	0.974	0.8835	0.9681	0.967
	PV2	0.912			
	PV3	0.908			
	PV4	0.964			
**Satisfaction**	SAT1	0.977	0.8699	0.9709	0.971
	SAT2	0.898			
	SAT3	0.907			
	SAT4	0.919			
	SAT5	0.96			
**Loyalty**	LOY1	0.961	0.8759	0.9724	0.972
	LOY2	0.911			
	LOY3	0.922			
	LOY4	0.918			
	LOY5	0.966			

**Table 7 behavsci-14-00973-t007:** Discriminant validity (Fornell–Larcker criterion).

	Convenience Value	Information Value	Experiential Value	Social Value	Perceived Risk	Perceived Cost	Perceived Value	Satisfaction	Loyalty
Convenience value	0.934								
Information value	0.532	0.934							
Experiential value	0.606	0.545	0.943						
Social value	0.482	0.476	0.543	0.942					
Perceived risk	−0.453	−0.477	−0.438	−0.511	0.936				
Perceived cost	−0.557	−0.494	−0.573	−0.504	0.544	0.930			
Perceived value	0.566	0.548	0.628	0.552	−0.541	−0.542	0.940		
Satisfaction	0.563	0.506	0.609	0.514	−0.467	−0.590	0.625	0.933	
Loyalty	0.525	0.539	0.542	0.460	−0.482	−0.509	0.531	0.553	0.936

**Table 8 behavsci-14-00973-t008:** Discriminant validity (HTMT values).

	Convenience Value	Information Value	Experiential Value	Social Value	Perceived Risk	Perceived Cost	Perceived Value	Satisfaction	Loyalty
Convenience value									
Information value	0.543								
Experiential value	0.616	0.543							
Social value	0.496	0.488	0.558						
Perceived risk	0.455	0.485	0.456	0.523					
Perceived cost	0.564	0.493	0.586	0.510	0.556				
Perceived value	0.583	0.557	0.640	0.564	0.554	0.559			
Satisfaction	0.582	0.515	0.620	0.525	0.487	0.600	0.642		
Loyalty	0.543	0.540	0.541	0.473	0.485	0.525	0.544	0.570	

**Table 9 behavsci-14-00973-t009:** Path coefficients for structural equation modeling.

Hypothesis	β	S.E.	C.R.	P	Results
H1a: Convenience value --> Perceived value	0.140	0.049	2.661	0.008 **	Supported
H1b: Information value --> Perceived value	0.138	0.054	2.797	0.005 **	Supported
H1c: Experiential value --> Perceived value	0.273	0.051	5.015	***	Supported
H1d: Social value --> Perceived value	0.145	0.052	2.931	0.003 **	Supported
H2a: Perceived risk --> Perceived value	−0.182	0.050	−3.742	***	Supported
H2b: Perceived cost --> Perceived value	−0.076	0.049	−1.442	0.149	Unsupported
H3: Perceived value --> Loyalty	0.320	0.058	5.513	***	Supported
H4: Perceived value --> Satisfaction	0.635	0.042	13.966	***	Supported
H5: Satisfaction --> Loyalty	0.350	0.063	5.988	***	Supported

Note: *** *p* < 0.001; ** *p* < 0.01.

**Table 10 behavsci-14-00973-t010:** Standardized total, direct, and indirect effects.

Perceived Value --> Satisfaction --> Customer Loyalty	Effect	Lower Bound	Upper Bound	Percentage
Total effect	0.542	0.450	0.625	
Direct effect	0.320	0.182	0.455	59%
Indirect effect	0.222	0.134	0.321	41%

## Data Availability

This article contains all data generated or analyzed during the study. The raw data can be obtained from the corresponding author upon reasonable request.

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
