# Peer review of "A Study on Customer Behavior in Online Dating Platforms: Analyzing the Impact of Perceived Value on Enhancing Customer Loyalty"

_behavsci, 2024, doi:10.3390/bs14100973_

Round 1
Reviewer 1 Report
Comments and Suggestions for Authors
Thank you for the opportunity to review this article. Overall, the research is well-executed and clearly written. However, my main concern pertains to the article's novelty. In marketing, value is typically defined as perceived customer value, which encompasses various types of benefits and costs—a well-established concept. Additionally, the relationships between value, customer satisfaction, and loyalty have been extensively explored. While the study successfully applies these relationships to the context of an online dating service, the findings primarily confirm rather than extend what is already widely known in the field. The authors can further beef up the contributions and novelty of this research in their General Discussion.
Author Response
For research article
|
Response to Reviewer 1Comments |
||
|
1. Summary |
|
|
|
Dear reviewer, Thank you very much for your detailed review and valuable comments on our paper, which are very helpful in making the paper more solid and fluent. We take each of your suggestions very seriously and have carefully considered and responded to each of them during the revision process. We have revised the manuscript carefully, and here are our responses to the comments and the corresponding revisions (comments in black, responses in blue, revisions in red) |
||
|
2. Questions for General Evaluation |
Reviewer’s Evaluation |
|
|
Is the content succinctly described and contextualized with respect to previous and present theoretical background and empirical research (if applicable) on the topic? |
Yes |
|
|
Are the research design, questions, hypotheses and methods clearly stated? |
Yes |
|
|
Are the arguments and discussion of findings coherent, balanced and compelling? |
Yes |
|
|
For empirical research, are the results clearly presented? |
Yes |
|
|
Is the article adequately referenced? |
Yes |
|
|
Are the conclusions thoroughly supported by the results presented in the article or referenced in secondary literature?
|
Yes |
|
|
3. Point-by-point response to Comments and Suggestions for Authors
|
||
|
Comments 1: While the study successfully applies these relationships to the context of an online dating service, the findings primarily confirm rather than extend what is already widely known in the field. The authors can further beef up the contributions and novelty of this research in their General Discussion.
|
||
|
Response 1: Thank you very much for your expert advice on our study. We realize that the elaboration of research contributions and novelty may not be sufficient in the previous version. Your feedback is very helpful for us to further improve this section. Based on your suggestions, we have delved into the Discussion, Contributions, Results, and Limitations sections of the paper to further emphasize the significance and novelty of this study and to ensure the completeness of the article's content and the academic contribution of the study. The specific changes are in lines 509-613 of the article.
Revision: 6. Discussion Second, this research explored the relationship among the six dimensions of perceived benefits, perceived sacrifices, and perceived value. The findings suggest that all components of perceived gain (convenience value, information value, experiential value, and social value) significantly influence perceived value, aligning with the findings of previous studies [11,68,69,123]. Notably, experiential value had the greatest impact. Fang et al. also highlighted in their study that experiential value is crucial for mobile platforms in sustaining a competitive advantage [89]. This suggests that experiential value is a prioritized motivation for customers of online dating platforms. This implies that a consistently high level in product and service quality has ceased to be a major factor in consumer choice [124]. Within the growing experiential economy, customers do not merely pay attention to the provider's offerings, but also to the memorable experiences and feelings during the usage process. Platforms can add a variety of interactive features, such as integrating entertainment content in the platform, such as games, live streaming, virtual gifts, etc., to make customer interactions on the platform richer and more interesting, and to increase the length of stay and stickiness of customers on the platform. Continuous innovation, such as the introduction of AR and VR technology, allows customers to experience dating in virtual reality, to enhance the customer's interactive experience. Social value has a positive impact on perceived value, which is consistent with the findings of Wang et al. [125,126]. As a dating and socializing application, online dating platforms uniquely combine the features of social media and matchmaking services. Users often seek to enhance their social image through these platforms, thereby achieving better outcomes in social interactions and dating experiences. To this end, future platforms can organize online or offline social events and themed activities, providing users with more opportunities to meet new people and engage in social interactions. By participating in discussions or events, users can gain more opportunities for self-presentation and recognition from others. Additionally, the platform could introduce a user rating or review system, where users earn points or titles through positive interactions, incentivizing them to maintain proactive behaviors. This approach would not only increase user engagement but also further enhance the platform's social value. Although convenience value has a positive impact on perceived value, it is not the most significant factor. This differs from the findings of Singh, S. et al., who concluded that convenience value plays a crucial role in influencing customer perceived value and has the greatest impact [68]. This difference may be due to changing times, where convenience is no longer a key factor in attracting customers but has instead become a basic requirement for online dating platforms. Additionally, information value has a significant impact on perceived value, indicating that relevant, sufficient, accurate, and timely information can enhance customer perceived value and encourage them to use online dating platforms. Therefore, in the future, providers should continue to improve platform convenience by optimizing user interfaces and enhancing the overall user experience. This includes ensuring a simple and user-friendly interface design that reduces operational complexity and allows users to communicate anytime, anywhere. Personalized recommendations and smarter matching algorithms can help users quickly find potential partners. Platforms should also ensure seamless functionality across devices (mobile phones, tablets, computers), allowing customers to easily switch between devices. Moreover, users should be encouraged to provide more detailed and accurate personal information. Enhancing the quality of information on the platform will enable customers to better understand potential matches without needing direct interaction. Finally, perceived risk exerts a significant adverse effect on perceived value, a finding that aligns with Chen, Q et al. conclusion that perceived risk significantly reduces customers' perceived value of online dating platforms [14]. This is because online dating platforms explicitly require customers to provide their cell phone numbers, personal photos, age, education, and other personal privacy information, which leads to customers' concerns about the risks associated with privacy leakage and being scammed, and therefore reduces perceived value [12,39]. While the role of perceived cost is relatively weak and the result is not significant, this outcome aligns with the findings of Zhong et al [11,65]. It may be due to the extensive use of smartphones and the advancement of infrastructure in recent years, which have significantly lowered the cost for customers to use online dating platforms, coupled with the fact that the competition in the online dating market is relatively fierce, and expenses such as access fees and membership charges are relatively small, and the perceived costs have a smaller influence on the perceived value of online dating platforms to some degree compared with customers in other mobile application markets. Therefore, platforms should actively encourage and ensure customer authenticity and integrity to reduce false information and fraudulent activities. By implementing strict customer verification and review mechanisms, platforms can ensure the authenticity and reliability of user profiles. To reduce perceived risk, platforms should strengthen privacy and data security protection measures, enhance user identity verification, and use encryption technologies to boost customers' trust in the platform. Additionally, improving the complaint and reporting mechanisms is essential to ensure that users can quickly re-port issues, and the platform can respond promptly to safeguard user rights.
7. Implications, Conclusions, and Limitations 7.1. Implications The theoretical contributions of this research are primarily demonstrated through the following aspects: First, this study is the first to apply the perceived value theory to the study of customer loyalty in online dating platforms, although the theory has been widely used in customer loyalty research in other fields [32,120,127]. Second, this study combines the perceived value theory with relevant literature [11,67,126] and divides perceived value into six dimensions based on the characteristics of online dating platforms: convenience value, information value, experience value, social value, perceived risk and perceived cost. By constructing a multi-dimensional framework of customer loyalty, this research uncovers the significant influence of perceived value on customer satisfaction and loyalty, further expanding the theoretical foundation of this field. In addition, the results of this study provide scholars with a direction for future research, showing that perceived value not only significantly affects customer satisfaction and loyalty, but also indirectly affects loyalty through satisfaction. Among them, experiential value, social value, informational value, convenience value, and perceived risk are important dimensions that affect perceived value, which provides an important breakthrough for future theoretical research on online dating platforms. In summary, this research not only offers an innovative theoretical framework for examining online dating platforms, but additionally establishes a robust basis for upcoming theoretical investigations. Attracting new users, retaining old users, and maintaining customer loyalty are crucial for online dating platforms [7]. From a practical perspective, this research offers several key suggestions for designers, administrators, and suppliers of online dating platforms. First, this study confirms the key role of perceived value in customer loyalty. Platform suppliers should continuously improve customer perceived value to deter users from migrating to rival platforms and achieve sustainable development. This is consistent with previous research [11,26]. Second, this study explores the impact of each value dimension on total perceived value, identifies the value components that are most important to customers, and helps platform designers develop effective strategies. Among them, experience value is the most valued dimension by customers, and earlier research has also indicated that experience value is crucial for platforms to maintain a competitive advantage [89]. As the social economy develops, individuals are paying more and more attention to meaningful experiences [128]. Therefore, plat-form providers should pay more attention to experience value and increase entertainment functions (such as games, live broadcasts, virtual gifts, etc.) to enhance customer loyalty. Simultaneously, perceived risk is also an important factor affecting customer loyalty. Platform designers and managers must comply with policies and regulations and implement effective information verification and privacy protection measures. By strengthening user authentication and information encryption, customer concerns about privacy leaks and fraud can be alleviated [14]. In summary, this study provides future directions for improving the practice of online dating platforms. It emphasizes the importance of improving customer experience and reducing perceived risk, which will help platform designers and managers develop more effective operational strategies.
|
||

Reviewer 2 Report
Comments and Suggestions for Authors
The research paper provides valuable insights into customer behavior in online dating platforms through the lens of perceived value. However, some issues need to be taken into consideration in order to enhance the quality of the paper.
Introduction
The introduction highlights the significance of customer loyalty in online dating platforms and briefly mentions gaps in existing literature. However, it lacks a thorough exploration of how online dating differs from other contexts where perceived value has been studied.
While perceived value is a central theory, other theories or frameworks that could complement or contrast with perceived value (e.g., technology acceptance models or social influence theories) are not discussed.
Literature Review
The review focuses primarily on perceived value but does not sufficiently critique the limitations or potential biases of applying this theory in the context of online dating platforms.
While some studies that are cited may have contradictory findings on perceived value, social value, and information value, these discrepancies are not fully explored.
Discussion
The discussion section tends to generalize the results without fully addressing the limitations of the study's scope and sample.
While the discussion offers some practical implications, it does not delve into specific, actionable strategies for online dating platform developers.
Conclusions and Limitations
Although the paper mentions some limitations, such as sample size and generalizability, it does not adequately address potential methodological biases or theoretical constraints.
The suggestions for future research are quite broad and lack specificity.
Some references are outdated or do not directly address the context of online dating platforms. The paper could benefit from including more recent studies that focus specifically on digital services and online user behavior. The following is suggested research
https://doi.org/10.1108/JFRA-12-2023-0765
Author Response
For research article
|
Response to Reviewer 2 Comments
|
||||||||||||||||||||||||||||||||||||||||||||||||||||||
|
1. Summary |
|
|
||||||||||||||||||||||||||||||||||||||||||||||||||||
|
Dear reviewer, Thank you very much for your detailed review and valuable comments on our paper, which are very helpful in making the paper more solid and fluent. We take each of your suggestions very seriously and have carefully considered and responded to each of them during the revision process. We have revised the manuscript carefully, and here are our responses to the comments and the corresponding revisions (comments in black, responses in blue, revisions in red):
|
||||||||||||||||||||||||||||||||||||||||||||||||||||||
|
2. Questions for General Evaluation |
Reviewer’s Evaluation |
|
||||||||||||||||||||||||||||||||||||||||||||||||||||
|
Is the content succinctly described and contextualized with respect to previous and present theoretical background and empirical research (if applicable) on the topic? |
Can be improved |
|
||||||||||||||||||||||||||||||||||||||||||||||||||||
|
Are the research design, questions, hypotheses and methods clearly stated? |
Can be improved |
|
||||||||||||||||||||||||||||||||||||||||||||||||||||
|
Are the arguments and discussion of findings coherent, balanced and compelling? |
Can be improved |
|
||||||||||||||||||||||||||||||||||||||||||||||||||||
|
For empirical research, are the results clearly presented? |
Yes |
|
||||||||||||||||||||||||||||||||||||||||||||||||||||
|
Is the article adequately referenced? |
Can be improved |
|
||||||||||||||||||||||||||||||||||||||||||||||||||||
|
Are the conclusions thoroughly supported by the results presented in the article or referenced in secondary literature?
|
Can be improved |
|
||||||||||||||||||||||||||||||||||||||||||||||||||||
|
3. Point-by-point response to Comments and Suggestions for Authors
|
||||||||||||||||||||||||||||||||||||||||||||||||||||||
|
Comments 1: The introduction highlights the significance of customer loyalty in online dating platforms and briefly mentions gaps in existing literature. However, it lacks a thorough exploration of how online dating differs from other contexts where perceived value has been studied.
|
||||||||||||||||||||||||||||||||||||||||||||||||||||||
|
Response 1: Thank you very much for your valuable comments on our study. Your feedback is greatly appreciated and has been extremely helpful to us. Based on your suggestions, we have revised the introduction section to provide a more detailed explanation of how online dating platforms differ from other contexts where perceived value has been studied. This addition aims to address the gap in the literature and enhance the clarity of our research focus. The specific revisions can be found in lines 61-84 of the manuscript. Once again, thank you for your insightful suggestions, which have significantly strengthened the introduction and overall coherence of our paper.
Revision: 1. Introduction Perceived value serves as a crucial predictor of customer loyalty and post-purchase attitudes and holds significant importance in the economic growth, survival, and competitiveness of a business [26,27]. In marketing, one of the primary tasks of a business is to provide value to customers and effectively communicate this value so as to boost customer satisfaction and loyalty, thereby enhancing profitability [28]. Perceived value stems from the consumer's overall assessment of the benefits and costs related to a product based on their evaluation of the product's utility [29]. If consumers perceive that the product or service is good value for money after comparing the benefits and sacrifices, they will be satisfied, thereby positively influencing loyalty [11]. The perceived value theory has been widely used in customer loyalty research in many fields, such as retail [30], e-commerce [26], digital marketing [31], mobile payments [32] and online transportation [33]. However, online dating platforms have unique characteristics, and the perceived value theory has not been fully explored in these research fields. Online dating platforms not only provide convenience value (e.g., the freedom to communicate anytime, anywhere) [34], information value (e.g., learning about potential partners through personal profiles) [35], experience value (e.g., exploring emotional and entertainment experiences) [36], and social value (e.g., gaining recognition through interactions) [37], but also involve a large amount of customers' privacy and personal information. Compared with other industries, customers' concerns about privacy breaches, fraud, and other issues make perceived risk particularly prominent on this platform [14]. This sensitivity and high relevance of risk is not commonly found in other research areas of perceived value. Therefore, applying the perceived value theory to online dating platforms, especially considering their unique risk factors and emotional interaction needs, is not only appropriate and necessary, but also effectively fills the research gaps in the existing literature.
Comments 2: While perceived value is a central theory, other theories or frameworks that could complement or contrast with perceived value (e.g., technology acceptance models or social influence theories) are not discussed.
Response 2: We sincerely appreciate your valuable feedback on our research. Your insights are incredibly helpful and have significantly contributed to improving our study, for which we are deeply grateful. Based on your suggestions, we have incorporated discussions of other relevant theories and frameworks (L. 509-613). The specific revisions are as follows:
Revision: 2.1. Online Dating Platforms Table 1. Review of the research literature on online dating platforms.
Comments 3: The review focuses primarily on perceived value but does not sufficiently critique the limitations or potential biases of applying this theory in the context of online dating platforms.
Response 3: Thank you very much for your expert opinion on our research. Your feedback is very important to us and points out important aspects that we have not been able to adequately address. We recognize that it is essential to critique the limitations and potential biases of applying the perceived value theory in the context of online dating platforms.
In response to your suggestion, we have added a discussion of the limitations and biases of the theory to the Limitations and Directions for Future Research of the revised manuscript. This addition is intended to provide a more comprehensive perspective and enhance the rigor of our analysis. The specific changes are in lines 650-656 of the article:
Revision: 7.3. Limitations and Future Research Directions This research offers fresh perspectives on the factors affecting customer loyalty to online dating platforms through the perceived value theory, but there are still some limitations. First, this study relies on self-disclosed information gathered through surveys, potentially leading to response biases and social desirability biases that impact the accuracy of the findings. Subsequent research could incorporate quantitative surveys and qualitative methods (such as interviews or focus groups) to gain more holistic and deeper understandings through mixed methods [12]. Second, this study's sample was limited to 352 customers, and the restricted sample size along with the geographical scope could influence the general applicability of the results. Subsequent studies ought to expand the sample size and increase geographical coverage to improve the comprehensiveness and representativeness of the research results [129]. In addition, due to resource and time constraints, this study used cross-sectional data, which partially limits the ability to monitor and interpret long-term changes in customer behavior and motivation on online dating platforms [130]. Subsequent research might expand our findings using a longitudinal approach to offer a more thorough understanding of the dynamic shifts in customer behavior. Finally, although the multi-dimensional model of perceived value theory is widely acknowledged as an effective tool, it may not sufficiently cover all the factors influencing customer loyalty to online dating platforms. Therefore, future research could consider combining other theoretical frameworks or introducing more variables, such as attachment theory and social influence theory, to more comprehensively understand the behavior of online dating platform customers [131]. This will help to further refine and expand on existing research results.
Comments 4: While some studies that are cited may have contradictory findings on perceived value, social value, and information value, these discrepancies are not fully explored.
Response 4: Thank you very much for your valuable comments on our study. Your feedback is extremely important to us and has greatly helped in improving the paper. We recognize that the previous manuscript did not adequately explore the differences between online dating platforms and other research contexts, particularly regarding the discrepancies in perceived value, social value, and information value as cited in the literature.
Based on your suggestions, we have expanded the literature review to further discuss the varying perspectives on the multidimensionality of perceived value and its classification standards in different studies. Additionally, we have provided a more detailed explanation of the specific characteristics of online dating platforms, such as convenience, information display, emotional experience, risks, and costs. Drawing from existing literature, we have categorized the perceived value of online dating platforms into two parts: the benefit aspect, which includes convenience value, information value, experiential value, and social value, and the sacrifice aspect, which includes perceived risks and costs. These revisions aim to more comprehensively reflect the application and uniqueness of perceived value in the context of online dating platforms.
The specific revisions can be found in lines 164-175 of the manuscript. Once again, thank you for your valuable suggestions. These adjustments have made our research model clearer and enhanced the depth and completeness of the literature review.
Revision: 2. Literature Review 2.2. Perceived value In numerous studies on the concept of value, scholars have recognized the multidimensional nature of value, but consensus has not been reached regarding how many dimensions exist or the classification standards [11,54]. Holbrook (1996) suggested that perceived value consists of eight dimensions: efficiency, recreation, excellence, aesthetics, status, moral respect, and spirituality, with each of these dimensions being interconnected [55]. Hsu et al. categorized perceived value into four dimensions: informational value, experiential value, social value, and transaction value [56]. In the context of mobile applications, Ling Jiang classified perceived value into social, informational, and hedonic dimensions [57]. Singh, S. et al. divided perceived value into convenience value, monetary value, social value, and emotional value in the context of streaming applications. Additionally, some scholars have suggested that perceived value can be distilled into two main dimensions: perceived benefit and perceived sacrifice. For instance, Bian et al. identified that perceived benefits encompass functional, social, emotional, cognitive, and experiential values, while perceived costs involve purchase expenses, purchase risks, and usage expenses [58]. Karsen et al. demonstrated that perceived risks and costs are crucial elements influencing the use of mobile applications [59,60]. Table 2 presents an overview of the prior research literature on perceived value. Online dating platforms provide customers with a convenient platform to communicate with potential partners anytime and anywhere. The platforms display detailed information about a large number of potential partners, and customers can get to know each other without interaction [34,35,61]. In addition, customers can also pursue fantasy, emotional and entertainment experiences on the platform[36], and gain recognition through active self-presentation and image building[37,62]. However, due to the lack of rigorous background checks and verification of personal information on the platform, fraud and illegal activities often occur [14]. At the same time, customers need to pay high membership subscription fees to use these services[63]. Based on the previous research, perceived value should include both benefits and sacrifices. Therefore, this study divides the benefits of perceived value into convenience value, information value, experience value, and social value, and the sacrifices into perceived risk and perceived cost.
Comments 5: The discussion section tends to generalize the results without fully addressing the limitations of the study's scope and sample.
Response 5: Thank you very much for your valuable comments on our study. In response to your suggestions. We have made the following modifications: 1. added how to avoid the limitations of the snowballing method in the Data Collection and Participant Demographics section of the article. 2. discussed in detail the specific limitations of the study scope and sample in the Limitations and Future Research Directions section of the article. We mentioned that the limitations of sample size and geographic coverage may affect the generalizability of the results and suggested that subsequent studies expand the sample size and geographic coverage.
The specific changes are in lines 358-362 and 641-645 of the article. Thank you again for your valuable comments.
Revision: 4.2. Data Collection and Participant Demographics This study employs a cross-sectional online questionnaire design, with data collected through China's professional online survey platform, Wenjuanxing (https://www.wjx.cn/). A questionnaire link and QR code were generated online, inviting users to complete the survey via social media platforms such as WeChat and QQ. Data collection utilized the snowball sampling technique across social media platforms [108]. Participants in the online questionnaire had the opportunity to enter a lottery, with prizes including:(1) a 5 RMB WeChat red packet;(2) a 10 RMB WeChat red packet;(3) a thank-you note for participation. In order to address the limitations of the snowball sampling method, we initially shared the questionnaire link within WeChat and QQ groups, avoiding an exclusive dependence on individual referrals. To ensure a more diverse sample, respondents were also requested to invite participants from various backgrounds, thereby reducing the risk of the sample being skewed toward a specific demographic.
7.3. Limitations and Future Research Directions This research offers fresh perspectives on the factors affecting customer loyalty to online dating platforms through the perceived value theory, but there are still some limitations. First, this study relies on self-disclosed information gathered through surveys, potentially leading to response biases and social desirability biases that impact the accuracy of the findings. Subsequent research could incorporate quantitative surveys and qualitative methods (such as interviews or focus groups) to gain more holistic and deeper understandings through mixed methods [12]. Second, this study's sample was limited to 352 customers, and the restricted sample size along with the geographical scope could influence the general applicability of the results. Subsequent studies ought to expand the sample size and increase geographical coverage to improve the comprehensiveness and representativeness of the research results [129]. In addition, due to resource and time constraints, this study used cross-sectional data, which partially limits the ability to monitor and interpret long-term changes in customer behavior and motivation on online dating platforms [130]. Subsequent research might expand our findings using a longitudinal approach to offer a more thorough understanding of the dynamic shifts in customer behavior. Finally, although the multi-dimensional model of perceived value theory is widely acknowledged as an effective tool, it may not sufficiently cover all the factors influencing customer loyalty to online dating platforms. Therefore, future research could consider combining other theoretical frameworks or introducing more variables, such as attachment theory and social influence theory, to more comprehensively understand the behavior of online dating platform customers [131]. This will help to further refine and expand on existing research results.
Comments 6: While the discussion offers some practical implications, it does not delve into specific, actionable strategies for online dating platform developers.
Response 6: Thank you very much for your valuable input on our research. Your feedback helped us realize that we did not delve into specific, actionable strategies for online dating platform developers in the Discussion section. Based on your suggestions, we have made the following revisions to the Discussion section to ensure that we are more specific in providing practical application recommendations for platform developers. 1. Enhance the user experience: We have introduced a number of actionable strategies to enhance the user experience. For example, adding interactive features such as games, live streaming, and virtual gifts to make interactions on the platform more interesting and extend user dwell time. Additionally, we recommend that platforms introduce AR and VR technologies to allow users to experience virtual dating to further enhance the interactive experience. 2. Enhancement of social value: We discuss in detail how to create more opportunities for users to interact with others by organizing online or offline social activities and motivate users to actively participate through a user rating system. Such measures can increase user engagement and at the same time enhance the social value of the platform. 3. Enhancement of information value and convenience: We suggest that the platform simplify operational complexity by optimizing the user interface, providing personalized recommendations and intelligent matching algorithms, and enabling users to communicate easily anytime, anywhere. These initiatives help improve the convenience and information value of the platform and enhance user satisfaction. 4. Reducing perceived risk: We have also proposed a series of measures to enhance privacy and data security, including strengthening user authentication, using encryption technology, and improving the complaint and reporting mechanism, in order to reduce users' concerns about privacy leakage and fraud. Through these initiatives, the platform can effectively enhance users' trust in the platform, thereby increasing the perceived value of users. These specific strategies are intended to provide practical guidance to online dating platform developers to help them make effective improvements in enhancing user experience and reducing perceived risk. The specific changes are in lines 502-570 of the article. Thank you again for your valuable suggestions that helped us improve the practical applicability of the paper and the depth of the discussion section.
Revision: 6. Discussion Second, this research explored the relationship among the six dimensions of perceived benefits, perceived sacrifices, and perceived value. The findings suggest that all components of perceived gain (convenience value, information value, experiential value, and social value) significantly influence perceived value, aligning with the findings of previous studies [11,68,69,123]. Notably, experiential value had the greatest impact. Fang et al. also highlighted in their study that experiential value is crucial for mobile platforms in sustaining a competitive advantage [89]. This suggests that experiential value is a prioritized motivation for customers of online dating platforms. This implies that a consistently high level in product and service quality has ceased to be a major factor in consumer choice [124]. Within the growing experiential economy, customers do not merely pay attention to the provider's offerings, but also to the memorable experiences and feelings during the usage process. Platforms can add a variety of interactive features, such as integrating entertainment content in the platform, such as games, live streaming, virtual gifts, etc., to make customer interactions on the platform richer and more interesting, and to increase the length of stay and stickiness of customers on the platform. Continuous innovation, such as the introduction of AR and VR technology, allows customers to experience dating in virtual reality, to enhance the customer's interactive experience. Social value has a positive impact on perceived value, which is consistent with the findings of Wang et al. [125,126]. As a dating and socializing application, online dating platforms uniquely combine the features of social media and matchmaking services. Users often seek to enhance their social image through these platforms, thereby achieving better outcomes in social interactions and dating experiences. To this end, future platforms can organize online or offline social events and themed activities, providing users with more opportunities to meet new people and engage in social interactions. By participating in discussions or events, users can gain more opportunities for self-presentation and recognition from others. Additionally, the platform could introduce a user rating or review system, where users earn points or titles through positive interactions, incentivizing them to maintain proactive behaviors. This approach would not only increase user engagement but also further enhance the platform's social value. Although convenience value has a positive impact on perceived value, it is not the most significant factor. This differs from the findings of Singh, S. et al., who concluded that convenience value plays a crucial role in influencing customer perceived value and has the greatest impact [68]. This difference may be due to changing times, where convenience is no longer a key factor in attracting customers but has instead become a basic requirement for online dating platforms. Additionally, information value has a significant impact on perceived value, indicating that relevant, sufficient, accurate, and timely information can enhance customer perceived value and encourage them to use online dating platforms. Therefore, in the future, providers should continue to improve platform convenience by optimizing user interfaces and enhancing the overall user experience. This includes ensuring a simple and user-friendly interface design that reduces operational complexity and allows users to communicate anytime, anywhere. Personalized recommendations and smarter matching algorithms can help users quickly find potential partners. Platforms should also ensure seamless functionality across devices (mobile phones, tablets, computers), allowing customers to easily switch between devices. Moreover, users should be encouraged to provide more detailed and accurate personal information. Enhancing the quality of information on the platform will enable customers to better understand potential matches without needing direct interaction. Finally, perceived risk exerts a significant adverse effect on perceived value, a finding that aligns with Chen, Q et al. conclusion that perceived risk significantly reduces customers' perceived value of online dating platforms [14]. This is because online dating platforms explicitly require customers to provide their cell phone numbers, personal photos, age, education, and other personal privacy information, which leads to customers' concerns about the risks associated with privacy leakage and being scammed, and therefore reduces perceived value [12,39]. While the role of perceived cost is relatively weak and the result is not significant, this outcome aligns with the findings of Zhong et al [11,65]. It may be due to the extensive use of smartphones and the advancement of infrastructure in recent years, which have significantly lowered the cost for customers to use online dating platforms, coupled with the fact that the competition in the online dating market is relatively fierce, and expenses such as access fees and membership charges are relatively small, and the perceived costs have a smaller influence on the perceived value of online dating platforms to some degree compared with customers in other mobile application markets. Therefore, platforms should actively encourage and ensure customer authenticity and integrity to reduce false information and fraudulent activities. By implementing strict customer verification and review mechanisms, platforms can ensure the authenticity and reliability of user profiles. To reduce perceived risk, platforms should strengthen privacy and data security protection measures, enhance user identity verification, and use encryption technologies to boost customers' trust in the platform. Additionally, improving the complaint and reporting mechanisms is essential to ensure that users can quickly report issues, and the platform can respond promptly to safeguard user rights.
Comments 7: Although the paper mentions some limitations, such as sample size and generalizability, it does not adequately address potential methodological biases or theoretical constraints.
Response 7: Thank you very much for your valuable comments on our study. Your feedback helped us further recognize the need for a more in-depth discussion of potential methodological biases and theoretical limitations beyond sample size and general applicability. Based on your suggestions, we have revised the limitations section with a new discussion of these issues. 1. Methodological bias: We realize that this study's reliance on survey-reported data may have led to response bias and social desirability bias, which in turn affected the accuracy of the results. Future research could combine quantitative questionnaires with qualitative methods (e.g., interviews or focus groups) to obtain more comprehensive insights through mixed methods. 2. Theoretical Limitations: Although the multidimensional model of perceived value theory is widely recognized as an effective tool for explaining user loyalty, it may not fully cover all the factors that affect user loyalty on online dating platforms. We include a discussion of other theoretical frameworks in our revision, particularly attachment theory and social influence theory, in order to explain user behavior and loyalty more comprehensively. Combining these theories allows for a better exploration of users' emotional attachments and the impact of social interactions on loyalty.
The specific revisions are in lines 637-656 in the article. With these revisions, we hope to respond more fully to your concerns about methodological biases and theoretical limitations.
Thank you again for your invaluable suggestions on our research; your feedback has helped us to significantly increase the depth and rigor of our research.
Revision: 7.3. Limitations and Future Research Directions This research offers fresh perspectives on the factors affecting customer loyalty to online dating platforms through the perceived value theory, but there are still some limitations. First, this study relies on self-disclosed information gathered through surveys, potentially leading to response biases and social desirability biases that impact the accuracy of the findings. Subsequent research could incorporate quantitative surveys and qualitative methods (such as interviews or focus groups) to gain more holistic and deeper understandings through mixed methods [12]. Second, this study's sample was limited to 352 customers, and the restricted sample size along with the geographical scope could influence the general applicability of the results. Subsequent studies ought to expand the sample size and increase geographical coverage to improve the comprehensiveness and representativeness of the research results [129]. In addition, due to resource and time constraints, this study used cross-sectional data, which partially limits the ability to monitor and interpret long-term changes in customer behavior and motivation on online dating platforms [130]. Subsequent research might expand our findings using a longitudinal approach to offer a more thorough understanding of the dynamic shifts in customer behavior. Finally, although the multi-dimensional model of perceived value theory is widely acknowledged as an effective tool, it may not sufficiently cover all the factors influencing customer loyalty to online dating platforms. Therefore, future research could consider combining other theoretical frameworks or introducing more variables, such as attachment theory and social influence theory, to more comprehensively understand the behavior of online dating platform customers [131]. This will help to further refine and expand on existing research results.
Comments 8: The suggestions for future research are quite broad and lack specificity.
Response 8: Thank you very much for your valuable comments on our study. Your feedback was very valuable and helpful and we appreciate it very much. Based on your suggestions, we have tailored our future research directions according to each limitation. The specific changes are in lines 635-656 of the article:
Revision: 7.3. Limitations and Future Research Directions This research offers fresh perspectives on the factors affecting customer loyalty to online dating platforms through the perceived value theory, but there are still some limitations. First, this study relies on self-disclosed information gathered through surveys, potentially leading to response biases and social desirability biases that impact the accuracy of the findings. Subsequent research could incorporate quantitative surveys and qualitative methods (such as interviews or focus groups) to gain more holistic and deeper understandings through mixed methods [12]. Second, this study's sample was limited to 352 customers, and the restricted sample size along with the geographical scope could influence the general applicability of the results. Subsequent studies ought to expand the sample size and increase geographical coverage to improve the comprehensiveness and representativeness of the research results [129]. In addition, due to resource and time constraints, this study used cross-sectional data, which partially limits the ability to monitor and interpret long-term changes in customer behavior and motivation on online dating platforms [130]. Subsequent research might expand our findings using a longitudinal approach to offer a more thorough understanding of the dynamic shifts in customer behavior. Finally, although the multi-dimensional model of perceived value theory is widely acknowledged as an effective tool, it may not sufficiently cover all the factors influencing customer loyalty to online dating platforms. Therefore, future research could consider combining other theoretical frameworks or introducing more variables, such as attachment theory and social influence theory, to more comprehensively understand the behavior of online dating platform customers [131]. This will help to further refine and expand on existing research results.
Comments 9: Some references are outdated or do not directly address the context of online dating platforms. The paper could benefit from including more recent studies that focus specifically on digital services and online user behavior.
Response 9: Thank you very much for your valuable input to our study. We really appreciate your feedback as it is very valuable and helpful to us. Based on your suggestions, we have updated the relevant literature.
|
||||||||||||||||||||||||||||||||||||||||||||||||||||||

Reviewer 3 Report
Comments and Suggestions for Authors
This manuscript utilises SEM to analyse the impact of 4 different dimensions of perceived customer and sacrifice to the overall evaluation of perceived value, satisfaction and loyalty. The discussion is holistic and based on the actual results. The research itself is interesting and new to the academia. Some suggestions are as follows:
It is OK that the abstract does not provide any trace of the China sample. But perhaps the authors can consider to include some clues of the research which is mainly drawn from a CHINA sample, and perhaps the social situation of dating in CHINA.
The second paragraph did not really bring out why a study to customer value and loyalty is needed.
The authors did not specify why these four dimensions of value in section 2.2 -3.1.4
The description for perceived risk in section 3.1.5 could further relate to the definition for perceived value (benefit and sacrifice)
Section 4.1 may further explain the original inventory / questionnaire.
In particular, for the perceived cost domain, it seems that Question #2 and #3 is different from the others. Would it be one the reason that led to the results, despite the satisfactory Cronbach Alpha value.
Minor suggestions:
There are 4 types of in-text citation from Line 127-132. Please unify the format for the whole manuscript.
Cross-sectional analysis (line 567) is not mentioned throughout the whole manuscript which is little bit confusing.
On line 588-591, perhaps it is important to comply with the requirement of the local government of both where the institution located at and the sample of where data is collected?
Author Response
For research article
|
Response to Reviewer 3 Comments
|
||||||||||||||||||||||||||||||||||||||||||||||||||||||||||||||||
|
1. Summary |
|
|
||||||||||||||||||||||||||||||||||||||||||||||||||||||||||||||
|
Dear reviewer, Thank you very much for your detailed review and valuable comments on our paper, which are very helpful in making the paper more solid and fluent. We take each of your suggestions very seriously and have carefully considered and responded to each of them during the revision process. We have revised the manuscript carefully, and here are our responses to the comments and the corresponding revisions (comments in black, responses in blue, revisions in red):
|
||||||||||||||||||||||||||||||||||||||||||||||||||||||||||||||||
|
2. Questions for General Evaluation |
Reviewer’s Evaluation |
|
||||||||||||||||||||||||||||||||||||||||||||||||||||||||||||||
|
Is the content succinctly described and contextualized with respect to previous and present theoretical background and empirical research (if applicable) on the topic? |
Can be improved |
|
||||||||||||||||||||||||||||||||||||||||||||||||||||||||||||||
|
Are the research design, questions, hypotheses and methods clearly stated? |
Can be improved |
|
||||||||||||||||||||||||||||||||||||||||||||||||||||||||||||||
|
Are the arguments and discussion of findings coherent, balanced and compelling? |
Yes |
|
||||||||||||||||||||||||||||||||||||||||||||||||||||||||||||||
|
For empirical research, are the results clearly presented? |
Yes |
|
||||||||||||||||||||||||||||||||||||||||||||||||||||||||||||||
|
Is the article adequately referenced? |
Can be improved |
|
||||||||||||||||||||||||||||||||||||||||||||||||||||||||||||||
|
Are the conclusions thoroughly supported by the results presented in the article or referenced in secondary literature? |
Can be improved |
|
||||||||||||||||||||||||||||||||||||||||||||||||||||||||||||||
|
3. Point-by-point response to Comments and Suggestions for Authors
|
||||||||||||||||||||||||||||||||||||||||||||||||||||||||||||||||
|
Comments 1: It is OK that the abstract does not provide any trace of the China sample. But perhaps the authors can consider to include some clues of the research which is mainly drawn from a CHINA sample, and perhaps the social situation of dating in CHINA.
|
||||||||||||||||||||||||||||||||||||||||||||||||||||||||||||||||
|
Response 1: Thank you very much for your valuable comments on our study. Based on your suggestions, we have made some revisions in the introduction section to highlight the relevance of the Chinese sample more clearly.
In the revised introduction, we emphasized the popularity of online dating platforms, illustrating the current state of the market. In addition, we specifically mention Momo's competitive position in the Chinese market as well as other major competitors such as TanTan and Baihe. This information is intended to give the reader a better understanding of the context and significance of the study.
The specific changes are in lines 37-40 in the article. Thank you again for your valuable suggestions; these changes significantly enhance the relevance and depth of our paper.
Revision: 1. Introduction The internet has transformed people's lifestyles, providing more opportunities for individuals to connect with others, expand their social networks, and find potential partners. Against this backdrop, online dating platforms have gained popularity among users [1]. These platforms can match users with prospective romantic partners and enable their initial interactions [2]. Reports indicate that nearly 40% of young individuals find their partners through online platforms, and by 2029, the global count of online dating users is projected to surpass 470 million [3]. As of June 2024, Tinder is the most downloaded dating app globally, with Bumble and Litmatch ranking second and third, respectively [4]. Despite its large customer base and rapid market growth, Tinder faces stiff competition in the market. As shown in Figure 1, there are several competitors offering similar services in the global market [4]. In the China market, Momo is the most popular dating app as of June 2024, but the company also faces fierce competition from many other competitors that provide similar services, such as TanTan, Baihe, iliao, Bumble, Qingteng Love, Soul, and other platforms [5]. In an increasingly competitive market environment, it has become common for customers to frequently switch between different platforms [6]. To meet this challenge, companies must not only strive to keep current customers, but also actively draw in new customers, with customer loyalty being vital to this process. [7]. Loyal customers usually continue to use the company's offerings, and through word-of-mouth communication, they can influence others, thereby helping companies acquire new customers at a lower cost [8,9]. Customer loyalty plays a key role in corporate profitability and sustainable development [10]. Loyal customers not only bring stable income to enterprises, but also help enterprises gain market share in competition and maintain long-term competitive advantages [11]. Comments 2: The second paragraph did not really bring out why a study to customer value and loyalty is needed. Response 2: Thank you for your valuable comments on our study. Based on your suggestions, we have revised the second paragraph to more clearly articulate the need for research on customer value and loyalty.
In the revised paragraph, we highlight that despite the high popularity of online dating platforms, there is a lack of research specific to the behavioral intent antecedents of these applications. We point out that existing research focuses on usage risk, psychobehavioral, demographic characteristics, and usage motivations, while the factors influencing customer loyalty remain unclear, indicating a significant research gap in this area.
In addition, we emphasize that understanding customer loyalty is critical for business survival and growth in the increasingly competitive online dating market. As the market environment changes, firms need to identify the key factors affecting customer loyalty and take steps to optimize the customer experience in order to retain existing customers and promote sustainable growth.
We also discuss the critical role of perceived value in customer loyalty and point out the lack of application of the theory in the specific domain of online dating platforms. This targeted discussion highlights the importance and usefulness of this study, which aims to provide valuable insights for designers and managers of online dating platforms.
Revision: 1. Introduction However, despite the fact that online dating platforms have the high prevalence, there are no specific studies explaining the antecedents of behavioral intentions to use such applications [12]. Research on online dating platforms has focused on risk of use [13–15], psychobehavior [16–18], homosexuality [19,20], sociodemographics [21,22], and motivations for use [23–25]. It is unclear which factors influence customer loyalty on online dating platforms, which is a research gap. As online dating continues to evolve, businesses need to determine the key factors affecting customer loyalty and take measures to optimize the customer experience so as to retain current customers, increase loyalty, and promote sustainable development of the business [9]. Perceived value serves as a crucial predictor of customer loyalty and post-purchase attitudes and holds significant importance in the economic growth, sur-vival, and competitiveness of a business [26,27]. In marketing, one of the primary tasks of a business is to provide value to customers and effectively communicate this value so as to boost customer satisfaction and loyalty, thereby enhancing profitability [28]. Perceived value stems from the consumer's overall assessment of the benefits and costs related to a product based on their evaluation of the product's utility [29]. If consumers perceive that the product or service is good value for money after comparing the benefits and sacrifices, they will be satisfied, thereby positively influencing loyalty [11]. The perceived value theory has been widely used in customer loyalty research in many fields, such as retail [30], e-commerce [26], digital marketing [31], mobile payments [32] and online transportation [33]. However, online dating platforms have unique char-acteristics, and the perceived value theory has not been fully explored in these research fields. Online dating platforms not only provide convenience value (e.g., the freedom to communicate anytime, anywhere) [34], information value (e.g., learning about po-tential partners through personal profiles) [35], experience value (e.g., exploring emo-tional and entertainment experiences) [36], and social value (e.g., gaining recognition through interactions) [37], but also involve a large amount of customers' privacy and personal information. Compared with other industries, customers' concerns about privacy breaches, fraud, and other issues make perceived risk particularly prominent on this platform [14]. This sensitivity and high relevance of risk is not commonly found in other research areas of perceived value. Therefore, applying the perceived value theory to online dating platforms, especially considering their unique risk factors and emotional interaction needs, is not only appropriate and necessary, but also effectively fills the research gaps in the existing literature. Therefore, this study constructed a conceptual model to explore online dating platform customer loyalty based on the perceived value theory by quantitatively analyzing 352 customers who have experienced online dating platforms, and using structural equation modeling (SEM) to test the relationship between the hypotheses. The findings assist online dating platform designers, managers, or vendors in identifying the specific elements influencing customer loyalty and in optimizing the customer experience so as to bolster market competitiveness and enhance customer loyalty. Comments 3: The authors did not specify why these four dimensions of value in section 2.2 -3.1.4 Response 3: Thank you very much for your valuable comments on our research. We realize that in the original manuscript, we did not adequately explain why these four value dimensions were emphasized. Your feedback is very helpful for us to improve this section. Based on your suggestions, we have revised and supplemented the relevant section to further explain the reasons for choosing these four value dimensions in order to enhance the persuasiveness of the theory and the completeness of the study.
The specific revisions can be viewed in lines 164-175 of the article:
Revision: 2.2. Perceived value Karsen et al. demonstrated that perceived risks and costs are crucial elements influencing the use of mobile applications [59,60]. Table 2 presents an overview of the prior research literature on perceived value. Online dating platforms provide customers with a convenient platform to communicate with potential partners anytime and anywhere. The platforms display de-tailed information about a large number of potential partners, and customers can get to know each other without interaction [34,35,61]. In addition, customers can also pursue fantasy, emotional and entertainment experiences on the platform [36], and gain recognition through active self-presentation and image building [37,62]. However, due to the lack of rigorous background checks and verification of personal information on the platform, fraud and illegal activities often occur [14]. At the same time, customers need to pay high membership subscription fees to use these services [63]. Based on the previous research, perceived value should include both benefits and sacrifices. Therefore, this study divides the benefits of perceived value into convenience value, information value, experience value, and social value, and the sacrifices into perceived risk and perceived cost. Comments 4: The description for perceived risk in section 3.1.5 could further relate to the definition for perceived value (benefit and sacrifice) Response 4: Thank you very much for your valuable feedback on our study. Your input has been incredibly helpful in improving the content of the paper. Based on your suggestion, we have revised the description of perceived risk and further connected it to the definition of perceived value (benefit and sacrifice). The revised content aims to better highlight the relationship between the two, thus strengthening the theoretical framework of the paper.
The specific revisions can be found in lines 262-280 of the manuscript:
Revision: 3.2.1. Perceived risk According to Bauer's definition, perceived risk is "the possibility that a consumer's behavior may have outcomes which are beyond his anticipation, some of which may be unpleasant [92]. Some researchers consider perceived risk as a sacrifice aspect of perceived value [11,87]. In our research, perceived risk denotes the likelihood and se-verity of negative outcomes resulting from the use of an online dating platform [14]. The matching algorithms used by online dating platforms not merely collect vast quantities of personal information but also autonomously gather data like location, time, and previous activities [93]. Such a wide range of online sources exposes online dating platforms to higher privacy risks compared to traditional offline dating services [14]. Second, deception through online dating platforms is very common due to the lack of strict background checks and profile verification, and potential fraud and other illicit activities frequently take place on online dating platforms [94]. In an online da-ting platform environment, customers may also receive numerous unsolicited communications and requests that could result in disruption of their lives and psychological stress [14]. Aware of these risks, people may develop a negative attitude towards online dating platforms, which in turn leads to a decrease in the perceived value of their services [87]. Thus, we suggest the following hypothesis: H2a. Perceived risk will negatively influence perceived value. Comments 5: Section 4.1 may further explain the original inventory / questionnaire. Response 5: Thank you very much for your valuable feedback on our study. Your input has been incredibly helpful. In the initial manuscript, the explanation of the questionnaire was not sufficiently detailed. Based on your suggestions, we have revised and expanded this section to provide a more thorough explanation of the original questionnaire, ensuring better clarity and understanding of its design and structure. The specific revisions can be found in lines 320-362 of the manuscript:
Revision: 4. Research Methodology 4.1. Questionnaire Development The gathering of questionnaire data comprised two components. The initial section gathers fundamental information regarding the customers, including gender, age, education level, and use of online dating platforms. The second section aligns with the research model and involves a total of nine variables: convenience value, information value, experiential value, social value, perceived risk, perceived cost, perceived value, satisfaction and loyalty. All variables were derived from previous literature to guarantee that the indicators were suitable for this study. Specifically, items for convenience value and information value were drawn from Ibáñez-Sánchez et al [56,65,67,68,101]. Items of experiential and social value were sourced from the re-search by Zhong et al. [11,66,91,102]. Items of perceived risk and perceived cost were drawn from the study by Chen et al. [11,12,14,59,103,104]. Items for perceived value were sourced from the research by Singh et al. [68,87,102]. Items for satisfaction were derived from the research by Hsu et al. [12,105,106], while items for loyalty refer to Yuan et al.'s study [11,101,106,107]. Each item was evaluated on a 7-point Likert scale, with options spanning from "1" for "strongly disagree" to "7" for "strongly agree." To ensure the accurate expression of the questionnaire, as the original questionnaire was in English and conducted in China, we invited three professional English translation researchers to translate and proofread the questionnaire several times be-fore it was distributed to reduce translation errors and eliminate ambiguities. Subsequently, we conducted a pretest of the questionnaire involving 30 customers with experience using online dating platforms to assess the logical flow and presentation of the questionnaire. In response to the researcher’s suggestions and feedback from the pretest, we modified the questionnaire. All respondents provided informed consent before completing the questionnaire to safeguard their rights and interests. Respond-ents were explicitly notified that the data collected would solely serve scholarly re-search purposes and that their privacy would be rigorously safeguarded. The list of entries and references of the questionnaire is provided in Table 3. 4.2. Data Collection and Participant Demographics This study employs a cross-sectional online questionnaire design, with data collected through China's professional online survey platform, Wenjuanxing (https://www.wjx.cn/). A questionnaire link and QR code were generated online, inviting users to complete the survey via social media platforms such as WeChat and QQ. Data collection utilized the snowball sampling technique across social media platforms [108]. Participants in the online questionnaire had the opportunity to enter a lottery, with prizes including:(1) a 5 RMB WeChat red packet;(2) a 10 RMB WeChat red packet;(3) a thank-you note for participation. In order to address the limitations of the snowball sampling method, we initially shared the questionnaire link within WeChat and QQ groups, avoiding an exclusive dependence on individual referrals. To ensure a more diverse sample, respondents were also requested to invite participants from various backgrounds, thereby reducing the risk of the sample being skewed to-ward a specific demographic. All participants had experience in using online dating platforms. Throughout the process, participants participated voluntarily and there was no conflict of interest. The researcher clearly communicated the study's purpose to the participants and emphasized the principle of data confidentiality. Participants were also informed that they could stop participating at any time if they felt uncomfortable. In this study, a total of 378 questionnaires were collected. All responses were meticulously reviewed, and invalid entries were excluded according to the following criteria: (1) answering all questions exactly the same; (2) completing the questionnaire in an excessively short period of time; and (3) questionnaires with apparently contradictory answers. Twenty-six questionnaires were finally determined to be invalid. Thus, 352 valid questionnaires were collected for this research. Among the respondents, 48% were male, while 52% were female. The respondents' ages were primarily concentrated in the 18-34 years old range (84.1%). 62.6% of the respondents had used online dating platforms for more than 6 months, and 64.2% of the respondents had used online dating platforms for more than 1 hour per day. These findings suggest that most respondents are inclined to use online dating platforms and are prepared to invest additional time. The descriptive analysis of the demographic information is shown in Table 4.
Comments 6: In particular, for the perceived cost domain, it seems that Question #2 and #3 is different from the others. Would it be one the reason that led to the results, despite the satisfactory Cronbach Alpha value.
Response 6: Thank you very much for your valuable comments on our study.We have noticed that questions 2 and 3 do differ from the other questions. After careful verification, we found that this was due to an error in the final manuscript preparation process, which resulted in other questions being incorrectly copied into the questionnaire. For this reason, we have thoroughly reviewed and proofread the questionnaire to ensure that the correct questions have been replaced and applied to the questionnaire.
Revision: 4.1. Questionnaire Development Table 3. Questionnaire Items.
Comments 7: There are 4 types of in-text citation from Line 127-132. Please unify the format for the whole manuscript. Response 7: Thank you very much for your valuable input to our study. We really appreciate your feedback. We have standardized the referencing of the texts. Comments 8: Cross-sectional analysis (line 567) is not mentioned throughout the whole manuscript which is little bit confusing. Response 8: Thank you very much for your valuable feedback on our study. Thank you very much for your feedback. We recognize that the original manuscript did not make sufficient reference to cross-sectional analysis, which may have led to some confusion. Following your suggestion, we have added a description of cross-sectional analysis in the questionnaire collection section and limitations section of the paper to provide a clearer explanation of cross-sectional analysis and to ensure that cross-sectional analysis is properly integrated throughout the paper.
See lines 350-352 and 645-648 of the manuscript for specific revisions. Revision: 4.2. Data Collection and Participant Demographics This study employs a cross-sectional online questionnaire design, with data collected through China's professional online survey platform, Wenjuanxing (https://www.wjx.cn/). A questionnaire link and QR code were generated online, inviting users to complete the survey via social media platforms such as WeChat and QQ. Data collection utilized the snowball sampling technique across social media platforms [108]. Participants in the online questionnaire had the opportunity to enter a lottery, with prizes including:(1) a 5 RMB WeChat red packet;(2) a 10 RMB WeChat red packet;(3) a thank-you note for participation. In order to address the limitations of the snowball sampling method, we initially shared the questionnaire link within WeChat and QQ groups, avoiding an exclusive dependence on individual referrals. To ensure a more diverse sample, respondents were also requested to invite participants from various backgrounds, thereby reducing the risk of the sample being skewed to-ward a specific demographic. 7.3. Limitations and Future Research Directions This research offers fresh perspectives on the factors affecting customer loyalty to online dating platforms through the perceived value theory, but there are still some limitations. First, this study relies on self-disclosed information gathered through surveys, potentially leading to response biases and social desirability biases that impact the accuracy of the findings. Subsequent research could incorporate quantitative surveys and qualitative methods (such as interviews or focus groups) to gain more holistic and deeper understandings through mixed methods [12]. Second, this study's sample was limited to 352 customers, and the restricted sample size along with the geo-graphical scope could influence the general applicability of the results. Subsequent studies ought to expand the sample size and increase geographical coverage to im-prove the comprehensiveness and representativeness of the research results [129]. In addition, due to resource and time constraints, this study used cross-sectional data, which partially limits the ability to monitor and interpret long-term changes in customer behavior and motivation on online dating platforms [130]. Subsequent research might expand our findings using a longitudinal approach to offer a more thorough understanding of the dynamic shifts in customer behavior. Finally, although the multidimensional model of perceived value theory is widely acknowledged as an effective tool, it may not sufficiently cover all the factors influencing customer loyalty to online dating platforms. Therefore, future research could consider combining other theoretical frameworks or introducing more variables, such as attachment theory and social influence theory, to more comprehensively understand the behavior of online dating platform customers [131]. This will help to further refine and expand on existing research results. Comments 9: On line 588-591, perhaps it is important to comply with the requirement of the local government of both where the institution located at and the sample of where data is collected? Response 9: Thank you very much for your valuable comments on our study. Your feedback is greatly appreciated. We recognize the importance of complying with the requirements of the local government, both where the institution is located and where the data is collected. Based on your suggestions, we have added this information and revised the relevant sections to ensure clarity and compliance.
The specific revisions can be found in lines 666-672 of the manuscript. Once again, thank you for your insightful suggestions, which have significantly contributed to improving the quality and rigor of our paper.
Revision: Institutional Review Board Statement: All participants provided informed consent before participating in the study. This study did not require ethical approval and complied with the local regulations of the institution's location(https://www.law.go.kr/LSW//lsLinkCommonInfo.do?lspttninfSeq=75929&chrClsCd=010202). Additionally, the study adhered to the local government requirements of the data collection site. According to Chapter III Ethical Review – Article 32 of the "Implementation of Ethical Review Measures for Human-Related Life Science and Medical Research" issued by the Chinese government, this study used anonymized information for research purposes, did not pose any harm to the subjects, and did not involve sensitive personal information or commercial interests; therefore, it was exempt from ethical review and approval(https://www.gov.cn/zhengce/zhengceku/2023-02/28/content_5743658.htm).
|
||||||||||||||||||||||||||||||||||||||||||||||||||||||||||||||||

Reviewer 4 Report
Comments and Suggestions for Authors The paper investigates a multidimensional perspective of perceived value to explore customer loyalty in online dating platforms and investigates the mediating role of satisfaction. The study employed cross-sectional data from a survey of 352 customers who had experienced online dating platforms and utilizing structural equation modeling (SEM) to examine the relationship among propositions, the research demonstrated a strong positive correlation between perceived value and satisfaction, loyalty, and a notable indirect impact on loyalty through satisfaction. Based on the paper provided for review, I propose recommendations to improve future versions of the paper.1-Introduction- The introduction section of the paper does not showcase the research gaps identified from recent literature. In Section 4.2, it is highlighted that the "Data for this study were gathered through Questionnaire Star (https://www.wjx.cn/), a specialized online survey platform in China". As a result, it is clear that this study is primarily concerned with online dating apps in China, as the entire survey population fell within this demographic. The introduction section provides key statistics on the overall growth of online dating platforms, but does not focus on the study's context. The authors highlight the growth of competitors in the global market for similar services, but do not explain how Chinese people use dating apps. Refer to the provided links and include more detailed statistics to support the research context. Links -Monthly active users of the largest mobile casual dating apps in China in 2024 -8 Most Popular Dating Apps in China 2-Literature Review- Section 2.1- The authors should consider creating a table that shows how various theoretical concepts have been linked to online dating platforms. 3-Hypothesis Development- The authors should consider adding more literature and arguments when proposing hypotheses (Section 3.2). Section 3.2- The figure 2 showcases a categorization of benefits and sacrifice in exploring the loyalty of online dating platforms. However, in the Section there are only a few lines (i.e., 3) of explanation regarding this categorization. Hence, please consider expanding your justification as it is a key element in your proposed model. 4- Research Methods- This title should be revised to research methodology. Moreover, the weakest segment of the paper lies in the methods section, which does not provide much information on how sampling was chosen and what type of sampling method was undertaken in the study. 5-Data Analysis- The authors should present HTMT results in addition to the Fornell and Larcker analyses to ensure discriminant validity. 6- Discussion- In the discussion section, a diagram of the accepted model should be presented along with the accepted hypothesis. 7- Implications, Conclusions, and Limitations- The authors should consider providing references to back up their arguments in this section. This section lacks support from current literature.
Author Response
Please find the file attached.

Reviewer 5 Report
Comments and Suggestions for Authors
Unfortunately, I think this study has very low differentiation and value.
The authors used the most frequently used variables in related previous studies without differentiation. Therefore, it is difficult to claim that this research model and research results are only applicable to online dating platforms.
What are the characteristics of the new platform called online dating, and why do the authors not reflect these characteristics in the research model?
The authors applied the variables used in previous studies by changing only the platform type. Does this have academic and practical implications?
Do the authors think that the results of the current study are different from previous studies? I don't think so at all.
Author Response
|
Response to Reviewer 5 Comments
|
||
|
1. Summary |
|
|
|
Dear reviewer, Thank you very much for your detailed review and valuable comments on our paper, which are very helpful in making the paper more solid and fluent. We take each of your suggestions very seriously and have carefully considered and responded to each of them during the revision process. We have revised the manuscript carefully, and here are our responses to the comments and the corresponding revisions (comments in black, responses in blue, revisions in red):
|
||
|
2. Questions for General Evaluation |
Reviewer’s Evaluation |
|
|
Is the content succinctly described and contextualized with respect to previous and present theoretical background and empirical research (if applicable) on the topic? |
Must be improved |
|
|
Are the research design, questions, hypotheses and methods clearly stated? |
Must be improved |
|
|
Are the arguments and discussion of findings coherent, balanced and compelling? |
Must be improved |
|
|
For empirical research, are the results clearly presented? |
Must be improved |
|
|
Is the article adequately referenced? |
Must be improved |
|
|
Are the conclusions thoroughly supported by the results presented in the article or referenced in secondary literature?
|
Must be improved |
|
|
3. Point-by-point response to Comments and Suggestions for Authors
|
||
|
Comments 1: What are the characteristics of the new platform called online dating, and why do the authors not reflect these characteristics in the research model?
|
||
|
Response 1: Thank you very much for your valuable comments on our study. Your feedback is greatly appreciated. We acknowledge that the characteristics of online dating platforms were not sufficiently detailed in our original manuscript, which may have limited the understanding of our research model. Based on your suggestion, we have revised the relevant sections to include a more comprehensive explanation of these characteristics and their relevance to the research model.
The specific revisions can be found in lines 164-176 of the manuscript. Once again, thank you for your insightful suggestions, which have significantly enhanced the clarity and depth of our paper.
Revision: 2.2. Perceived value Online dating platforms provide customers with a convenient platform to communicate with potential partners anytime and anywhere. The platforms display detailed information about a large number of potential partners, and customers can get to know each other without interaction [34,35,61]. In addition, customers can also pursue fantasy, emotional and entertainment experiences on the platform [36], and gain recognition through active self-presentation and image building [37,62]. However, due to the lack of rigorous background checks and verification of personal information on the platform, fraud and illegal activities often occur [14]. At the same time, customers need to pay high membership subscription fees to use these services [63]. Based on the previous research, perceived value should include both benefits and sacrifices. Therefore, this study divides the benefits of perceived value into convenience value, information value, experience value, and social value, and the sacrifices into perceived risk and perceived cost.
Comments 2: The authors applied the variables used in previous studies by changing only the platform type. Does this have academic and practical implications? Response 2: Thank you very much for your insightful and constructive feedback on our study. We deeply appreciate your comments, which have been extremely valuable in helping us improve our work. In response to your suggestion, we have elaborated further on both the academic and practical implications of our study, particularly with regard to the modification of platform types while utilizing variables from previous studies. We recognize the importance of addressing this aspect, and we have accordingly revised the relevant sections (L. 572-613) to provide a more comprehensive explanation. Once again, thank you for your thoughtful review, which has contributed significantly to enhancing the quality of our paper.
Revision: 7.1. Implications The theoretical contributions of this research are primarily demonstrated through the following aspects: First, this study is the first to apply the perceived value theory to the study of customer loyalty in online dating platforms, although the theory has been widely used in customer loyalty research in other fields [32,120,127]. Second, this study combines the perceived value theory with relevant literature [11,67,126] and divides perceived value into six dimensions based on the characteristics of online dating platforms: convenience value, information value, experience value, social value, perceived risk and perceived cost. By constructing a multi-dimensional framework of customer loyalty, this research uncovers the significant influence of perceived value on customer satisfaction and loyalty, further expanding the theoretical foundation of this field. In addition, the results of this study provide scholars with a direction for future research, showing that perceived value not only significantly affects customer satisfaction and loyalty, but also indirectly affects loyalty through satisfaction. Among them, experiential value, social value, informational value, convenience value, and perceived risk are important dimensions that affect perceived value, which provides an important breakthrough for future theoretical research on online dating platforms. In summary, this research not only offers an innovative theoretical framework for examining online dating platforms, but additionally establishes a robust basis for upcoming theoretical investigations. Attracting new users, retaining old users, and maintaining customer loyalty are crucial for online dating platforms [7]. From a practical perspective, this research offers several key suggestions for designers, administrators, and suppliers of online dating platforms. First, this study confirms the key role of perceived value in customer loyalty. Platform suppliers should continuously improve customer perceived value to deter users from migrating to rival platforms and achieve sustainable development. This is consistent with previous research [11,26]. Second, this study explores the impact of each value dimension on total perceived value, identifies the value components that are most important to customers, and helps platform designers develop effective strategies. Among them, experience value is the most valued dimension by customers, and earlier research has also indicated that experience value is crucial for platforms to maintain a competitive advantage [89]. As the social economy develops, individuals are paying more and more attention to meaningful experiences [128]. Therefore, plat-form providers should pay more attention to experience value and increase entertainment functions (such as games, live broadcasts, virtual gifts, etc.) to enhance customer loyalty. Simultaneously, perceived risk is also an important factor affecting customer loyalty. Platform designers and managers must comply with policies and regulations and implement effective information verification and privacy protection measures. By strengthening user authentication and information encryption, customer concerns about privacy leaks and fraud can be alleviated [14]. In summary, this study provides future directions for improving the practice of online dating platforms. It emphasizes the importance of improving customer experience and reducing perceived risk, which will help platform designers and managers develop more effective operational strategies.
Comments 3: Do the authors think that the results of the current study are different from previous studies? I don't think so at all. Response 3: Thank you very much for your valuable comments on our study. We deeply appreciate your feedback, which has been extremely helpful in improving our work. In response to your suggestion, we have carefully reviewed the comparison between the results of our study and previous research. We have made adjustments to the discussion section (L. 509-570) to further explore the similarities and differences between our findings and those of previous studies. We recognize the importance of thoroughly addressing this point and have clarified it more explicitly in the revised version.
Once again, thank you for your valuable insights, which have provided important guidance for improving our research.
Revision: 6. Discussion Second, this research explored the relationship among the six dimensions of perceived benefits, perceived sacrifices, and perceived value. The findings suggest that all components of perceived gain (convenience value, information value, experiential value, and social value) significantly influence perceived value, aligning with the findings of previous studies [11,68,69,123]. Notably, experiential value had the greatest impact. Fang et al. also highlighted in their study that experiential value is crucial for mobile platforms in sustaining a competitive advantage [89]. This suggests that experiential value is a prioritized motivation for customers of online dating platforms. This implies that a consistently high level in product and service quality has ceased to be a major factor in consumer choice [124]. Within the growing experiential economy, customers do not merely pay attention to the provider's offerings, but also to the memorable experiences and feelings during the usage process. Platforms can add a variety of interactive features, such as integrating entertainment content in the platform, such as games, live streaming, virtual gifts, etc., to make customer interactions on the platform richer and more interesting, and to increase the length of stay and stickiness of customers on the platform. Continuous innovation, such as the introduction of AR and VR technology, allows customers to experience dating in virtual reality, to enhance the customer's interactive experience. Social value has a positive impact on perceived value, which is consistent with the findings of Wang et al. [125,126]. As a dating and socializing application, online dating platforms uniquely combine the features of social media and matchmaking services. Users often seek to enhance their social image through these platforms, thereby achieving better outcomes in social interactions and dating experiences. To this end, future platforms can organize online or offline social events and themed activities, providing users with more opportunities to meet new people and engage in social interactions. By participating in discussions or events, users can gain more opportunities for self-presentation and recognition from others. Additionally, the platform could introduce a user rating or review system, where users earn points or titles through positive interactions, incentivizing them to maintain proactive behaviors. This approach would not only increase user engagement but also further enhance the platform's social value. Although convenience value has a positive impact on perceived value, it is not the most significant factor. This differs from the findings of Singh, S. et al., who concluded that convenience value plays a crucial role in influencing customer perceived value and has the greatest impact [68]. This difference may be due to changing times, where convenience is no longer a key factor in attracting customers but has instead become a basic requirement for online dating platforms. Additionally, information value has a significant impact on perceived value, indicating that relevant, sufficient, accurate, and timely information can enhance customer perceived value and encourage them to use online dating platforms. Therefore, in the future, providers should continue to improve platform convenience by optimizing user interfaces and enhancing the overall user experience. This includes ensuring a simple and user-friendly interface design that reduces operational complexity and allows users to communicate anytime, anywhere. Personalized recommendations and smarter matching algorithms can help users quickly find potential partners. Platforms should also ensure seamless functionality across devices (mobile phones, tablets, computers), allowing customers to easily switch between devices. Moreover, users should be encouraged to provide more detailed and accurate personal information. Enhancing the quality of information on the platform will enable customers to better understand potential matches without needing direct interaction. Finally, perceived risk exerts a significant adverse effect on perceived value, a finding that aligns with Chen, Q et al. conclusion that perceived risk significantly reduces customers' perceived value of online dating platforms [14]. This is because online dating platforms explicitly require customers to provide their cell phone numbers, personal photos, age, education, and other personal privacy information, which leads to customers' concerns about the risks associated with privacy leakage and being scammed, and therefore reduces perceived value [12,39]. While the role of perceived cost is relatively weak and the result is not significant, this outcome aligns with the findings of Zhong et al [11,65]. It may be due to the extensive use of smartphones and the advancement of infrastructure in recent years, which have significantly lowered the cost for customers to use online dating platforms, coupled with the fact that the competition in the online dating market is relatively fierce, and expenses such as access fees and membership charges are relatively small, and the perceived costs have a smaller influence on the perceived value of online dating platforms to some degree compared with customers in other mobile application markets. Therefore, platforms should actively encourage and ensure customer authenticity and integrity to reduce false information and fraudulent activities. By implementing strict customer verification and review mechanisms, platforms can ensure the authenticity and reliability of user profiles. To reduce perceived risk, platforms should strengthen privacy and data security protection measures, enhance user identity verification, and use encryption technologies to boost customers' trust in the platform. Additionally, improving the complaint and reporting mechanisms is essential to ensure that users can quickly re-port issues, and the platform can respond promptly to safeguard user rights.
|
||

Round 2
Reviewer 2 Report
Comments and Suggestions for Authors
Thank you for your thoughtful revisions and for addressing the reviewers' comments so thoroughly. I appreciate the effort you've put into improving the manuscript, and I'm pleased to recommend it for publication in its present form.
Author Response
Dear reviewers,
Thank you very much for your acknowledgement and positive feedback on our revised manuscript. We greatly appreciate your valuable suggestions and guidance throughout the review process, which played an important role in improving the quality of the paper.
Thank you again for your time and support.
Reviewer 3 Report
Comments and Suggestions for Authors
Perhaps the authors can still take an even more detailed look on the in-text citation. Also problems are found in the reference page.
Extract from 149-154
Holbrook (1996) suggested that perceived value consists of eight dimensions: efficiency, recreation, excellence, aesthetics, status, moral respect, and spirituality, with each of these dimensions being interconnected [55].
Hsu et al. categorized perceived value into four dimensions: informational value, experiential value, social value, and transaction value [56]. In the context of mobile applications,
Ling Jiang classified perceived value into social, informational, and hedonic dimensions [57].
Author Response
For research article
|
Response to Reviewer 3 Comments
|
||
|
1. Summary |
|
|
|
Dear reviewer, Thank you very much for your detailed review and valuable comments on our paper, which are very helpful in making the paper more solid and fluent. We take each of your suggestions very seriously and have carefully considered and responded to each of them during the revision process. We have revised the manuscript carefully, and here are our responses to the comments and the corresponding revisions (comments in black, responses in blue, revisions in red):
|
||
|
2. Questions for General Evaluation |
Reviewer’s Evaluation |
|
|
Is the content succinctly described and contextualized with respect to previous and present theoretical background and empirical research (if applicable) on the topic? |
Can be improved |
|
|
Are the research design, questions, hypotheses and methods clearly stated? |
Yes |
|
|
Are the arguments and discussion of findings coherent, balanced and compelling? |
Yes |
|
|
For empirical research, are the results clearly presented? |
Yes |
|
|
Is the article adequately referenced? |
Can be improved |
|
|
Are the conclusions thoroughly supported by the results presented in the article or referenced in secondary literature? |
Yes |
|
|
3. Point-by-point response to Comments and Suggestions for Authors
|
||
|
Comments 1: Perhaps the authors can still take an even more detailed look on the in-text citation. Also problems are found in the reference page. Extract from 149-154 Holbrook (1996) suggested that perceived value consists of eight dimensions: efficiency, recreation, excellence, aesthetics, status, moral respect, and spirituality, with each of these dimensions being interconnected [55].Hsu et al. categorized perceived value into four dimensions: informational value, experiential value, social value, and transaction value [56]. In the context of mobile applications,Ling Jiang classified perceived value into social, informational, and hedonic dimensions [57]. |
||
|
Response 1: Thank you very much for your valuable feedback. We appreciate your suggestions and have made the following revisions to address the concerns regarding the in-text citations and reference page: 1.In-text citations: We have reviewed and revised the in-text citations to ensure consistency and accuracy throughout the manuscript. We have corrected the formatting of the citations to match the required style, ensuring they are presented appropriately according to the journal's guidelines. 2.Reference page: Upon your recommendation, we have carefully examined the reference page and corrected any errors in formatting, ordering, and completeness. We have ensured that all references correspond accurately to the in-text citations and are presented in a consistent format, following the journal's requirements. We hope these revisions meet your expectations. Thank you again for your insightful comments, which have helped us improve the quality of the paper.
Revision: Extract from 147-152 In numerous studies on the concept of value, scholars have recognized the multidimensional nature of value, but consensus has not been reached regarding how many dimensions exist or the classification standards [11,53]. Holbrook suggested that perceived value consists of eight dimensions: efficiency, recreation, excellence, aesthetics, status, moral respect, and spirituality, with each of these dimensions being interconnected [54]. Hsu et al. categorized perceived value into four dimensions: informational value, experiential value, social value, and transaction value [55]. In the context of mobile applications, Jiang et al. classified perceived value into social, informational, and hedonic dimensions [56]. Singh et al. divided perceived value into convenience value, monetary value, social value, and emotional value in the context of streaming applications. Additionally, some scholars have suggested that perceived value can be distilled into two main dimensions: perceived benefit and perceived sacrifice. For instance, Bian et al. identified that perceived benefits encompass functional, social, emotional, cognitive, and experiential values, while perceived costs involve purchase expenses, purchase risks, and usage expenses [57]. Karsen et al. demonstrated that perceived risks and costs are crucial elements influencing the use of mobile applications [58,59]. Table 2 presents an overview of the prior research literature on perceived value.
|
||

Reviewer 4 Report
Comments and Suggestions for Authors
First and foremost, I would like to thank the authors for revising the manuscript and bringing it to a good level. However, I believe there is a minor scope for improvement in the manuscript. Hence, it would be fantastic if the authors could make the following comments:
1- Literature Review- The table is presented effectively, highlighting a variety of studies. However, it would be beneficial to highlight the findings of the study, which used a theoretical lens to conduct its research (i.e., rather than stating purpose of the study). 2- Research Model- The authors should consider changing the title to "Conceptual Framework" and attempting to keep it consistent in Figure 2 with the similar figure title "Proposed Conceptual Framework". 3- The authors have stated about the hedonic and utilitarian needs in Mobile application Sections 3.2.1. However, try to seek ideas from the paper below and include the dimension of perceived risk and costs, as well as cite - Acharya, N. and Sharma, A., 2022, June. Demarcating the privacy issues of Aarogya Setu app in Covid-19 pandemic in India: an exploration into contact tracing mobile applications from elaboration likelihood model. In International Conference on Human-Computer Interaction (pp. 457-468). Cham: Springer International Publishing. 3- Questionnaire Development- The authors should add a statement on whether these 30 customers that a part of pilot study were later included as a participant in primary analysis after final questionnaire development or where excluded as they belong to the pilot study only. 4- Were questionnaire items reversed to ensure face validity? If so, please clarify in Section 4.1. 5- The authors should include a notes section below Table 9 stating: - Notes: *p<.05; **p<.01, ***p<.001Author Response
For research article
|
Response to Reviewer 4 Comments
|
||||||||||||||||||||||||||||||||||||||||||||||||||||||||||||||||||||||||||||||||||||||||||||||||||||||||||||||||||||||||||||||||||||||||||||||||||||||||||||||||||||||||||||||||||||||||||||||||||||||||||||||||||||||||||||||||||||||||||||
|
1. Summary |
|
|
||||||||||||||||||||||||||||||||||||||||||||||||||||||||||||||||||||||||||||||||||||||||||||||||||||||||||||||||||||||||||||||||||||||||||||||||||||||||||||||||||||||||||||||||||||||||||||||||||||||||||||||||||||||||||||||||||||||||||
|
Dear reviewer, Thank you very much for your detailed review and valuable comments on our paper, which are very helpful in making the paper more solid and fluent. We take each of your suggestions very seriously and have carefully considered and responded to each of them during the revision process. We have revised the manuscript carefully, and here are our responses to the comments and the corresponding revisions (comments in black, responses in blue, revisions in red):
|
||||||||||||||||||||||||||||||||||||||||||||||||||||||||||||||||||||||||||||||||||||||||||||||||||||||||||||||||||||||||||||||||||||||||||||||||||||||||||||||||||||||||||||||||||||||||||||||||||||||||||||||||||||||||||||||||||||||||||||
|
2. Questions for General Evaluation |
Reviewer’s Evaluation |
|
||||||||||||||||||||||||||||||||||||||||||||||||||||||||||||||||||||||||||||||||||||||||||||||||||||||||||||||||||||||||||||||||||||||||||||||||||||||||||||||||||||||||||||||||||||||||||||||||||||||||||||||||||||||||||||||||||||||||||
|
Is the content succinctly described and contextualized with respect to previous and present theoretical background and empirical research (if applicable) on the topic? |
Yes |
|
||||||||||||||||||||||||||||||||||||||||||||||||||||||||||||||||||||||||||||||||||||||||||||||||||||||||||||||||||||||||||||||||||||||||||||||||||||||||||||||||||||||||||||||||||||||||||||||||||||||||||||||||||||||||||||||||||||||||||
|
Are the research design, questions, hypotheses and methods clearly stated? |
Can be improved |
|
||||||||||||||||||||||||||||||||||||||||||||||||||||||||||||||||||||||||||||||||||||||||||||||||||||||||||||||||||||||||||||||||||||||||||||||||||||||||||||||||||||||||||||||||||||||||||||||||||||||||||||||||||||||||||||||||||||||||||
|
Are the arguments and discussion of findings coherent, balanced and compelling? |
Yes |
|
||||||||||||||||||||||||||||||||||||||||||||||||||||||||||||||||||||||||||||||||||||||||||||||||||||||||||||||||||||||||||||||||||||||||||||||||||||||||||||||||||||||||||||||||||||||||||||||||||||||||||||||||||||||||||||||||||||||||||
|
For empirical research, are the results clearly presented? |
Yes |
|
||||||||||||||||||||||||||||||||||||||||||||||||||||||||||||||||||||||||||||||||||||||||||||||||||||||||||||||||||||||||||||||||||||||||||||||||||||||||||||||||||||||||||||||||||||||||||||||||||||||||||||||||||||||||||||||||||||||||||
|
Is the article adequately referenced? |
Yes |
|
||||||||||||||||||||||||||||||||||||||||||||||||||||||||||||||||||||||||||||||||||||||||||||||||||||||||||||||||||||||||||||||||||||||||||||||||||||||||||||||||||||||||||||||||||||||||||||||||||||||||||||||||||||||||||||||||||||||||||
|
Are the conclusions thoroughly supported by the results presented in the article or referenced in secondary literature?
|
Yes |
|
||||||||||||||||||||||||||||||||||||||||||||||||||||||||||||||||||||||||||||||||||||||||||||||||||||||||||||||||||||||||||||||||||||||||||||||||||||||||||||||||||||||||||||||||||||||||||||||||||||||||||||||||||||||||||||||||||||||||||
|
3. Point-by-point response to Comments and Suggestions for Authors
|
||||||||||||||||||||||||||||||||||||||||||||||||||||||||||||||||||||||||||||||||||||||||||||||||||||||||||||||||||||||||||||||||||||||||||||||||||||||||||||||||||||||||||||||||||||||||||||||||||||||||||||||||||||||||||||||||||||||||||||
|
Comments 1: Literature Review- The table is presented effectively, highlighting a variety of studies. However, it would be beneficial to highlight the findings of the study, which used a theoretical lens to conduct its research (i.e., rather than stating purpose of the study).
|
||||||||||||||||||||||||||||||||||||||||||||||||||||||||||||||||||||||||||||||||||||||||||||||||||||||||||||||||||||||||||||||||||||||||||||||||||||||||||||||||||||||||||||||||||||||||||||||||||||||||||||||||||||||||||||||||||||||||||||
|
Response 1: Thank you for your valuable comments on our literature review section. Based on your suggestions, we have revised the table to highlight the research findings using a theoretical perspective, rather than just stating the research objectives. We believe these changes will make the table more clearly demonstrate the core findings of the research and further enhance the depth and practicality of the literature review. Thank you again for your constructive comments, which have helped us improve the quality of the article.The specific changes are in the table on lines 131-132 of the article: Revision: Table 1. Review of the research literature on online dating platforms.
Comments 2:Research Model- The authors should consider changing the title to "Conceptual Framework" and attempting to keep it consistent in Figure 2 with the similar figure title "Proposed Conceptual Framework". Response 2: Thank you very much for your suggestion regarding the research model section. Based on your comment, we have changed the title to ‘Conceptual model’ and ensured consistency with the title of Figure 2, ‘Proposed conceptual model’. We believe this revision improves the overall consistency and clarity of the article. Thank you again for your valuable suggestions, which have helped us further improve the structure of the paper.For specific changes, see lines 307 of the article: Revision: 3.4. Conceptual model Exploring the loyalty of online dating platform customers is important for the development, optimisation and promotion of the platform. This research develops a thorough model grounded in the multidimensional perspective of perceived value (Figure 2). According to the preceding discussion, the framework covers both the perceived benefits and perceived sacrifices components of perceived value, contains six dimensions, and considers satisfaction as an important component affecting loyalty. We aim to offer theoretical backing and empirical evidence for the design and marketing strategy of online dating platforms. Figure 2 illustrates the hypothesised relationships between the variables: Figure 2. Proposed conceptual model. Comments 3:The authors have stated about the hedonic and utilitarian needs in Mobile application Sections 3.2.1. However, try to seek ideas from the paper below and include the dimension of perceived risk and costs, as well as cite - Acharya, N. and Sharma, A., 2022, June. Demarcating the privacy issues of Aarogya Setu app in Covid-19 pandemic in India: an exploration into contact tracing mobile applications from elaboration likelihood model. In International Conference on Human-Computer Interaction (pp. 457-468). Cham: Springer International Publishing. 3- Questionnaire Development- The authors should add a statement on whether these 30 customers that a part of pilot study were later included as a participant in primary analysis after final questionnaire development or where excluded as they belong to the pilot study only. Response 3: Thank you very much for your valuable feedback on our paper. Based on your suggestions, we have made the following revisions and additions: 1.According to your suggestion, we have referenced and cited the paper you mentioned in section 3.2.1. Perceived Risk: 2.Regarding questionnaire development, we have added a clarification in the text, explicitly stating that the 30 customers who participated in the pilot study were not included in the main analysis after the final questionnaire development, and we have provided an explanation for their exclusion. We believe this addition further enhances the transparency and rigor of the research. Thank you again for your valuable suggestions, which have helped us improve the content and structure of the paper. The specific changes can be found in lines 265-268,340-344 of the revised manuscript. Revision:3.2.1. Perceived risk According to Bauer's definition, perceived risk is "the possibility that a consumer's behavior may have outcomes which are beyond his anticipation, some of which may be unpleasant" [91]. Some researchers consider perceived risk as a sacrifice aspect of perceived value [11,86]. In our research, perceived risk denotes the likelihood and severity of negative outcomes resulting from the use of an online dating platform [14]. The matching algorithms used by online dating platforms not merely collect vast quantities of personal information but also autonomously gather data like location, time, and previous activities [92,93]. Such a wide range of online sources exposes online dating platforms to higher privacy risks compared to traditional offline dating services [14]. Second, deception through online dating platforms is very common due to the lack of strict background checks and profile verification, and potential fraud and other illicit activities frequently take place on online dating platforms [94]. In an online dating platform environment, customers may also receive numerous unsolicited communications and requests that could result in disruption of their lives and psychological stress [14]. Aware of these risks, people may develop a negative attitude towards online dating platforms, which in turn leads to a decrease in the perceived value of their services [86]. Thus, we suggest the following hypothesis: H2a. Perceived risk will negatively influence perceived value. 4.1. Questionnaire Development To ensure the accurate expression of the questionnaire, as the original questionnaire was in English and conducted in China, we invited three professional English translation researchers to translate and proofread the questionnaire several times before it was distributed to reduce translation errors and eliminate ambiguities. Subsequently, we conducted a pretest of the questionnaire involving 30 customers with experience using online dating platforms to assess the logical flow and presentation of the questionnaire. Based on the researcher’s suggestions and feedback from the pretest, we modified the questionnaire to improve its clarity and structure. The 30 customers who participated in the pretest were excluded from the primary analysis to avoid potential bias arising from their prior exposure to the questionnaire. All respondents provided informed consent before completing the questionnaire to safeguard their rights and interests. Respondents were explicitly notified that the data collected would solely serve scholarly research purposes and that their privacy would be rigorously safeguarded. The list of entries and references of the questionnaire is provided in Table 3.
Comments 4: Were questionnaire items reversed to ensure face validity? If so, please clarify in Section 4.1.
Response 4: Thank you for your valuable comments on our article. Based on your suggestions, we have corrected the order of the questionnaire questions in section 4.1. Thank you again for your suggestions, which have helped us improve the rigor of the paper. The specific changes are in the table on lines 422–423 in the article:
Revision:
Comments 5: The authors should include a notes section below Table 9 stating: - Notes: *p<.05; **p<.01, ***p<.001
Response 5: Thank you for your valuable comments on our article. Based on your suggestions, we have added a notes section below Table 9, which reads as follows: Note: ***p < 0.001; **p < 0.01; *p < 0.05.We believe this addition makes the statistical information in the table clearer and easier to understand. Thank you again for your suggestions, which have helped us further improve the quality of the article. For specific changes, see lines 460 of the article: Revision:Table 9. Path coefficients for structural equation modelling.
Note: ***p < 0.001; **p < 0.01; *p < 0.05.
|
||||||||||||||||||||||||||||||||||||||||||||||||||||||||||||||||||||||||||||||||||||||||||||||||||||||||||||||||||||||||||||||||||||||||||||||||||||||||||||||||||||||||||||||||||||||||||||||||||||||||||||||||||||||||||||||||||||||||||||

Reviewer 5 Report
Comments and Suggestions for Authors
I acknowledge that the quality of the research has significantly improved due to the authors' efforts. However, I believe there are still issues with the research model originally proposed by the authors. The model primarily reuses variables that are frequently employed in previous studies, failing to introduce new value. Moreover, the unique characteristics of online dating platforms should be reflected in the research model to yield novel insights, but the current model does not do this effectively. Therefore, despite the authors' sincere efforts, I have concluded that the paper is not yet at a level suitable for publication in this journal.
Author Response
Dear reviewers,
First of all, I would like to express my sincere gratitude for your valuable feedback on my manuscript. Each of your comments has been crucial in improving my research. I would like to provide further clarification regarding your suggestions in the hope of gaining your
understanding.
As you pointed out, we indeed utilized some variables that are commonly found in other studies. These variables have been widely applied in the field of customer loyalty. Our objectiveis
to test their applicability in the unique context of online dating platforms and, through their
combination, enrich the existing research. Although these variables have been employed in other
studies, their roles may differ in various contexts. Our research aims to explore the specific
impact of these variables on customer behavior within the online dating platform environment.
We fully agree that the unique characteristics of online dating platforms should be reflected in the research. This is why we have specifically introduced perceived value as a core variable. Perceived value plays a crucial role in this study for several reasons: it reflects them ultidimensional nature of user experience, encompassing not only functional aspects but also emotional interaction and social needs, which are particularly important in an environment where social interaction is highly dependent, like online dating platforms. Perceived value is a key driver of customer loyalty, as users who obtain high-quality emotional and social interaction experiences are more likely to develop emotional attachment and loyalty to the platform. Additionally, perceived value highlights the differentiation strategies of online dating platforms, as they enhance user value through personalized services and privacy protection, thereby better attracting and retaining users. Therefore, perceived value not only helps us capture the uniqueness of online dating platforms but also provides new insights into the mechanisms that shape customer loyalty.
We are willing to incorporate more innovative variables into the research model based on your valuable suggestions in future studies, with the aim of achieving higher academic value.
Once again, thank you for your time and guidance throughout the review process.
